# O-Edit: Orthogonal Subspace Editing for Language Model Sequential Editing

## Abstract

Large language models (LLMs) acquire knowledge during pre-training, but over time, this knowledge may become incorrect or outdated, necessitating updates after training. Knowledge editing techniques address this issue without the need for costly re-training. However, most existing methods are designed for single edits, and as the number of edits increases, they often cause a decline in the model's overall performance, posing significant challenges for sequential editing. To overcome this, we propose Orthogonal Subspace Editing, O-Edit. This algorithm orthogonalizes the direction of each knowledge update, minimizing interference between successive updates and reducing the impact of new updates on unrelated knowledge. Our approach does not require replaying previously edited data and processes each edit knowledge on time. It can perform thousands of edits on mainstream LLMs, achieving an average performance improvement that is 4.2 times better than existing methods while effectively preserving the model's performance on downstream tasks, all with minimal additional parameter overhead.

## 1 Introduction

Large language models (LLMs) are trained on vast amounts of textual data, enabling them to store extensive knowledge about various aspects of the human world, sparking the potential for general artificial intelligence. However, LLMs face significant challenges, including the propagation of inaccurate or outdated knowledge, as well as the generation of bias or harmful content (Cai et al., 2024b; Chen et al., 2024; Zhong et al., 2024). Given the substantial computational costs of re-training LLMs to address these issues, there has been growing interest in model editing techniques (Yao et al., 2023; Wang et al., 2023a), which aim to update specific content within the model while minimizing computational costs. Existing model editing methods can be categorized into two main types: parameter-modifying methods that directly alter a small subset of model parameters (Dai et al., 2022; Meng et al., 2023a;b; Hu et al., 2024a;b; Gupta et al., 2024a), and parameter-preserving methods that without changing the model parameters (Wang et al., 2024b; Cai et al., 2024a; Zheng et al., 2023). In this paper, we focus on parameter-modifying editing methods.

Most existing research focuses on editing models a single time (Han et al., 2023; Zhang et al., 2024b;a; Mazzia et al., 2024). However, as real-world knowledge continuously evolves, models will need to be updated repeatedly to remain accurate. This shift has led to the concept of sequential model editing (Ma et al., 2024; Hu et al., 2024b; Huang et al., 2023), which involves performing multiple knowledge edits to progressively update the model as new knowledge needs to be incorporated. Currently, sequential editing is often achieved through multiple iterations of single edits. Recent studies have shown that as the number of edits increases, the success rate of edits significantly declines and impairs the model's general capabilities, such as reasoning and contextual understanding, thereby limiting the scalability of model editing (Gu et al., 2024; Gupta et al., 2024a;b). This challenge is akin to adding new floors to an existing building—each addition risks compromising the overall stability. While some research has analyzed the bottlenecks of sequential editing from a theoretical perspective (Ma et al., 2024; Hu et al., 2024a), there is still no effective solution has yet been developed to address this issue through direct modifications of the model weights.[1].

---

[1] For more details on related work, please refer to Appendix A.

To address the scalability issue of sequential editing, this paper introduces Orthogonal Subspace Editing (O-Edit), a simple yet effective method for sequentially editing language models. Our key insight is based on the observation that existing editing methods primarily perform updates within specific low-rank subspaces. Based on this premise, we assume that both the update directions from previous editing tasks and the directions of updates to the model's implicit knowledge can be captured. Therefore, for the current editing knowledge, the direction of parameter updates should be chosen to minimize the impact on these prior update directions. O-Edit accomplishes this by projecting the update direction of the current knowledge into an orthogonal subspace, ensuring that the neural network's output for previous knowledge remains unchanged while the projected direction remains effective for the current edit. To enhance O-Edit, we introduce O-Edit+, a post-processing method designed to ensure complete orthogonality between subspaces. We validate the effectiveness of our methods by utilizing two knowledge editing datasets and four downstream task datasets. Furthermore, our analysis, conducted from both experimental and theoretical perspectives, clearly demonstrates that strong orthogonality between each update matrix is crucial for enabling sequential editing. Figure 1 illustrates how our methods adjust the update direction for each piece of knowledge.

Our method offers four key advantages: (1) **Efficiency**: It requires minimal additional parameters while enabling hundreds or even thousands of sequential edits. (2) **Privacy**: There is no requirement to store the edited data itself, ensuring privacy during updates. (3) **Timeliness**: Our method allows for the immediate application of each edit, making it more practical. (4) **Flexibility**: Our method is compatible with existing sequential editing techniques, allowing for easy integration and adaptability to various scenarios.

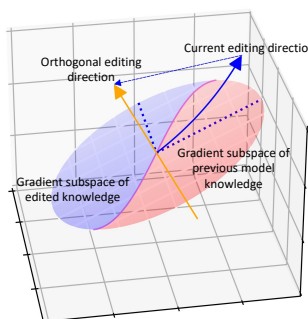

Figure 1: O-Edit constrains the direction of each update to lie within an orthogonal subspace.

Our main contributions are as follows: ① We introduce O-Edit and O-Edit+, two simple and efficient methods for sequential editing in large language models (LLMs) that can handle thousands of edits in orthogonal subspaces, effectively addressing the performance degradation issue encountered by existing approaches during multiple edits. ② Our methods significantly preserve model performance on downstream tasks, demonstrating their scalability and practicality even after numerous sequential edits in real-world continuous model update scenarios. ③ We show that the orthogonality between knowledge is essential for supporting sequential editing, providing a viable research direction for this task.

## 2 PRELIMINARIES

In this section, we introduce sequential model editing. Subsequently, in Section 3, we discuss two prominent knowledge editing techniques, ROME (Meng et al., 2023a) and MEMIT (Meng et al., 2023b), and extend them into the sequential editing method O-Edit. Finally, in Section 4, we further refine O-Edit by presenting O-Edit+, a more straightforward and effective approach for orthogonal sequential model editing.

We focus on the challenge of sequential model editing (SME) (Wang et al., 2024b; Ma et al., 2024), which aims to enable large language models (LLMs) to undergo extensive sequential modifications, potentially involving hundreds or thousands of edits. The primary objective is to ensure that the model's outputs align with human expectations across target queries, while simultaneously preserving the LLM's pre-existing knowledge and capabilities. Let $f_\Theta : \mathbb{X} \to \mathbb{Y}$, parameterized by $\Theta$, denote a model function that maps an input $\mathbf{x}$ to its corresponding prediction $f_\Theta(\mathbf{x})$. The initial model, $f_{\Theta_0}$, is pre-trained on a large dataset $D_{\text{train}}$. When the LLM exhibits inaccuracies or requires updates, model editing becomes necessary, using a dynamic, time-evolving dataset $\mathcal{D}_{\text{edit}} = \{(\mathcal{X}_e, \mathcal{Y}_e) \mid (x_1, y_1), \ldots, (x_T, y_T)\}$. At each time step $T$, a model editor (ME) applies the $T$-th edit, updating the previous model $f_{\Theta_{T-1}}$ to produce a new model $f_{\Theta_T}$, following the equation:

$$f_{\Theta_T} = \text{ME}(f_{\Theta_{T-1}}, \mathbf{x}_T, y_T), \quad \text{s.t.} \quad f_{\Theta_T}(\mathbf{x}) = \begin{cases} y_T & \text{if } \mathbf{x} \in \mathcal{X}_e, \\ f_{\Theta_0}(\mathbf{x}) & \text{if } \mathbf{x} \notin \mathcal{X}_e. \end{cases} \quad (1)$$

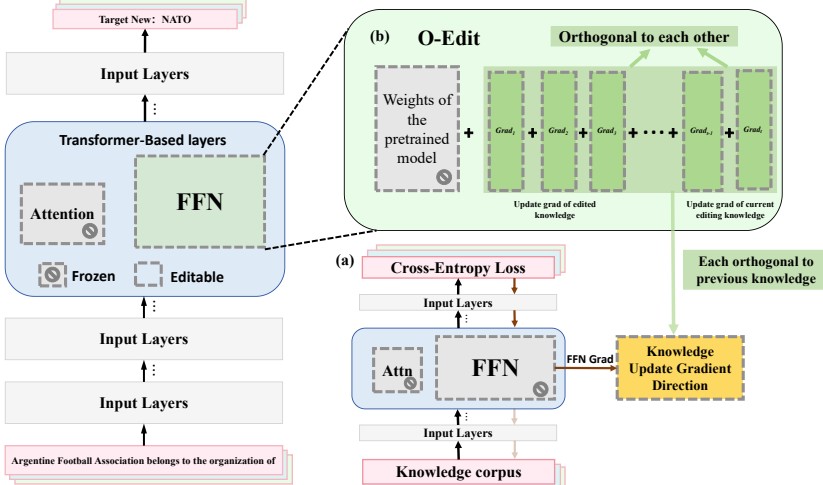

Figure 2: The framework of O-Edit for sequential language model editing. **(a)** First, we compute gradients on a large amount of textual data without updating the model parameters. This step provides the gradient information necessary for updating model's implicit knowledge. **(b)** Next, we impose constraints on the update directions for each piece of edited knowledge, ensuring these directions are orthogonal to each other as well as to the directions of the model's implicit knowledge.

Eqn. 1 indicates that after model editing, the LLM should correctly predict the current edit with $f_{\Theta_T}(\mathbf{x}_T) = y_T$, while preserving previous edits $(\mathbf{x}_{<T}, y_{<T}) \in \mathcal{D}_{\text{edit}}$ are inaccessible to the editor, the model is still able to retain this edit. Additionally, the model should maintain the performance of the original model $f_{\Theta_0}$ on data outside the editing scope, $\mathbf{x} \notin \mathcal{X}_e$, particularly with respect to the general training corpus $D_{\text{train}}$.

# 3  **O-EDIT**: SEQUENTIAL EDITING WITH GRADIENT PROJECTION MEMORY

In this section, we will introduce O-Edit, as illustrated in Figure 2. We discuss two key-value memory-based knowledge editing methods, ROME and MEMIT in Appendix B.3 and B.4, followed by our optimization method in section 3.1, which incrementally edits new knowledge in orthogonal subspaces, while preserving previously edited knowledge.

## 3.1  TOWARDS AN ORTHOGONAL EDITING METHOD

Previous methods share a common feature: all new knowledge is updated within a shared space, which directly affects the weights of the model. If an update for new knowledge is applied without considering prior knowledge, the direction of this update can affect both the previously edited knowledge and the implicit knowledge within the model, potentially leading to catastrophic forgetting (Luo et al., 2024; Wang et al., 2023c). Therefore, to effectively support sequential editing, the process of updating new knowledge should adhere to the following criteria:

**Criterion 3.1:** The update direction for each piece of knowledge should be orthogonal to the directions of previously edited knowledge, ensuring minimal interference with previously edited knowledge.

**Criterion 3.2:** The update direction for each piece of knowledge should be orthogonal to the implicit knowledge directions within the original model, ensuring minimal interference with the model's existing implicit knowledge.

In the following sections 3.1.1 and 3.1.2, we will detail how we optimized ROME and MEMIT to fulfill the two criteria mentioned above within the context of sequential editing.

### 3.1.1  THE KNOWLEDGE TO BE EDITED SHOULD BE MUTUALLY ORTHOGONAL

**Editing the First Piece of Knowledge:** To comply with criterion 3.1, we implement the following steps in a sequential editing process. We commence by editing the first piece of knowledge

using the pair $(x_1, y_1)$. Upon completion of this initial edit, we obtain an updated set of parameters $\Delta W_{[\text{total}]} = \Delta W_{[1]}$. To preserve this edited knowledge, we constrain the gradient update directions for subsequent edits. It is important to note that during the editing process with methods such as ROME and MEMIT, parameter adjustments are made without gradient computation, as the calculation of $v_*$ necessitates training, while the adjustment of $W_{proj}$ occurs in a single step. Since ROME and MEMIT do not involve computing the gradient direction of the required update matrix, we draw on the insights from (Wang et al., 2023b) and utilize $\Delta W_{[\text{total}]}$ to approximate the direction of model parameter updates. They argue that the gradient space from prior training tasks can be effectively captured by the update matrix. Next, we perform Singular Value Decomposition (SVD) on $\Delta W_{[\text{total}]} = U\Sigma V^T$ and extract the sub-matrix $\Delta W_r$ corresponding to the top $r$ singular values, defined as the Core Gradient Space (CGS) by (Saha et al., 2021). Updates along the CGS direction induce maximum changes in knowledge (Farajtabar et al., 2019), whereas updates in directions orthogonal to the CGS minimize interference with previously edited knowledge[2].

**Editing the Subsequent Knowledge:** To edit the second piece of knowledge using examples from $D_{\text{edit}}$, we first retrieve the bases of the Core Gradient Space (CGS). The new update direction must lie in the space orthogonal to the CGS:

$$\Delta W_r^T \cdot \Delta W_{[2]} = \mathbf{0}. \tag{2}$$

This ensures that the column vector subspace of $W_2$ is orthogonal to the column vector subspace of $W_r$. Taking MEMIT as an example, the update in Eq.18 can be optimized as[3]:

$$\widetilde{W} = W + (v_* - Wk_*)k_*^T(C + k_*k_*^T)^{-1},$$
$$\text{where} \quad \Delta W_r^T \cdot (v_* - Wk_*)k_*^T(C + k_*k_*^T)^{-1} = \mathbf{0}. \tag{3}$$

Non-trivial solutions that approximately satisfy Eqn.3 can be obtained by training $v_*$, where Eqn.17 can be rewritten as:

$$\mathcal{L}(v) + \lambda_1 f_1(\Delta W_r; v). \tag{4}$$

Here:

$$f_1 = \text{sim}\left(\Delta W_r, (v_* - Wk_*)k_*^T(C + k_*k_*^T)^{-1}\right), \tag{5}$$

**sim** represents the cosine similarity function in column vector space, where each column vector lies in $\mathbb{R}^d$, and $\lambda_1$ serves as a hyperparameter that regulates the degree of orthogonality. Upon completion of the training of $v_*$, Eqn. 18 is employed to determine the update parameter $\Delta W_{[2]}$. Following the update of the second piece of knowledge, the edited parameters are revised as follows:

$$\Delta W_{[\text{total}]} + = \Delta W_{[2]}. \tag{6}$$

We then proceed to the next piece of new knowledge, repeating the same procedure as for the second piece. The value of $r$ increases linearly with each iteration of knowledge editing, defined as $r = \min(1 \times \text{Iteration}, \text{rank}(\Delta W_{[\text{total}]}))$. We provide an efficient solution for Eqn. 5 and an explanation for $r$ in Appendix B.5.

### 3.1.2 THE EDITED KNOWLEDGE SHOULD BE ORTHOGONAL TO THE IMPLICIT KNOWLEDGE

To adhere to criterion 3.2, we implement the following steps in the sequential editing process. We perform backpropagation on a large corpus of text to capture the model's gradient information for the update direction of its internal implicit knowledge while freezing the original model's (unedited) parameters, simulating the pre-training process without updating the model, as illustrated in the bottom right of Figure 2. This computation is conducted on Wikipedia text, accumulating the gradient information by summing it. Appendix B.6 provides a comparison for selecting the appropriate text. Notably, this involves actual gradient information rather than the approximate update direction used in Section 3.1.1.

---

[2]For additional details on updating within orthogonal subspaces, please refer to Appendix A.3 and B.1.

[3]Since Eqn.15 involves **matrix right multiplication**, $d$ denotes the column dimension and $d_m$ denotes the row dimension.

Once the gradient information $\nabla G \in \mathbb{R}^{d \times d_m}$ of the implicit knowledge is obtained, the update direction for knowledge editing should be orthogonal to $\nabla G$. Similar to Section 3.1.1, we obtain the rank $q$ approximation of $\nabla G$, denoted as $\nabla G_q$, through SVD. We then subtract the projection of $\nabla G_q$ onto $W_r$ from $\nabla G_q$:

$$\nabla G_q = \nabla G_q - \Delta W_r (\Delta W_r^T \Delta W_r)^{-1} \Delta W_r^T \nabla G_q, \tag{7}$$

to prevent knowledge conflicts (Xu et al., 2024; Jin et al., 2024) between the two. For instance, if $\Delta W_r$ contains the edited knowledge *"The President of the US is Harris/Trump"*, while $\nabla G_q$ contains *"The President of the US is Biden"*, the update directions for these two pieces of knowledge may conflict or even be completely opposite. In such cases, we prioritize preserving the knowledge in $\Delta W_r$ over $\nabla G_q$. The ultimate training objective is:

$$\text{loss} = \mathcal{L}(z) + \lambda_1 f_1(\Delta W_r; v) + \lambda_2 f_2(\nabla G_q; v), \tag{8}$$

where:

$$f_2 = \text{sim}\left(\nabla G_q, (v_* - W k_*) k_*^T (C + k_* k_*^T)^{-1}\right). \tag{9}$$

The rank $q$ increases linearly with the number of iterations of knowledge editing, described by $q = \lambda_3 \times \text{iteration}$, where $\lambda_3$ is a hyperparameter controlling the degree of constraints.

Eqn. 8 represents the final optimization target. After obtaining $v_*$, we use Eqn. 18 to solve for the update parameter. We then update the hyperparameters $r$, $q$, and $\Delta W_{[\text{total}]}$ for the next knowledge update.

## 4  O-EDIT+: TOWARDS MORE EFFICIENT SEQUENTIAL MODEL EDITING

In Section 3, we introduced O-Edit, an algorithm for approximate orthogonal sequential knowledge editing. To further enhance the orthogonality between different pieces of knowledge, we propose O-Edit+, a post-processing method that eliminates the need for cosine similarity calculations. Specifically, for the second piece of knowledge, we compute $v_*$ using Eqn.17 and apply Eqn.18 to obtain the update parameter $\Delta W_{[2]}$. Subsequently, $\Delta W_{[2]}$ undergoes post-orthogonal processing, achieved as follows:

$$\Delta W_{[2]} = \Delta W_{[2]} - \Delta W_r (\Delta W_r^T \Delta W_r)^{-1} \Delta W_r^T \Delta W_{[2]},$$
$$\nabla G_q = \nabla G_q - \Delta W_r (\Delta W_r^T \Delta W_r)^{-1} \Delta W_r^T \nabla G_q, \tag{10}$$
$$\Delta W_{[2]} = \Delta W_{[2]} - \nabla G_q (\nabla G_q^T \nabla G_q)^{-1} \nabla G_q^T \Delta W_{[2]}.$$

The processed $\Delta W_{[2]}$ from Eqn.10 is then used as the update direction for the second piece of knowledge. Similar to O-Edit, we subsequently update the hyperparameters $r$, $q$, and $\Delta W_{[\text{total}]}$ for the next knowledge edit. We detail the computation process of Eqn.10 and the pseudo-code for O-Edit and O-Edit+ in Appendix B.5. Readers can refer to Appendices B.8 and B.9 for details on hyperparameter selection.

## 5  EXPERIMENTS

### 5.1  EDITING EXPERIMENTAL SETTINGS AND EVALUATION METRICS

**Datasets and Models.** We utilize autoregressive LLMs, specifically **Mistral-7B** (Jiang et al., 2023) and **Llama3-8B**[4], for evaluation, along with the datasets **ZsRE** (Cao et al., 2021), **COUNTER-FACT** (Meng et al., 2023a), **RECENT** and **WIKICF** (Zhang et al., 2024a).

**Baseline.** We selected **Fine-Tuning (FT)** (Yao et al., 2023), **FT-EWC** (Wang et al., 2024b), **MEND** (Mitchell et al., 2022a), **ROME** (Meng et al., 2023a) and **MEMIT** (Meng et al., 2023b) as baseline editors and compared them with our proposed methods, **O-Edit**, **O-Edit+** and ♠ **O-Edit+** which

---

[4]https://llama.meta.com/llama3

Table 1: **Main editing results for COUNTERFACT.** $T$: Num Edits.

| Method | COUNTERFACT | | | | | | | | | | | | | | | |
| | $T = 200$ | | | | $T = 500$ | | | | $T = 1000$ | | | | $T = 1500$ | | | |
| | Rel. | Gen. | Loc. | Avg. | Rel. | Gen. | Loc. | Avg. | Rel. | Gen. | Loc. | Avg. | Rel. | Gen. | Loc. | Avg. |
| **Mistral-7B** | | | | | | | | | | | | | | | | |
| FT | 0.31 | 0.12 | 0.19 | 0.21 | 0.09 | 0.03 | 0.02 | 0.04 | 0.05 | 0.01 | 0.01 | 0.03 | 0.03 | 0.01 | 0.00 | 0.01 |
| FT-EWC | 0.68 | 0.34 | 0.22 | 0.41 | 0.26 | 0.17 | 0.10 | 0.17 | 0.12 | 0.05 | 0.09 | 0.09 | 0.09 | 0.04 | 0.07 | 0.07 |
| MEND | 0.51 | 0.22 | 0.21 | 0.31 | 0.19 | 0.09 | 0.07 | 0.12 | 0.12 | 0.03 | 0.02 | 0.05 | 0.07 | 0.02 | 0.01 | 0.03 |
| ROME | 0.72 | 0.53 | 0.31 | 0.52 | 0.30 | 0.18 | 0.14 | 0.21 | 0.28 | 0.10 | 0.06 | 0.15 | 0.27 | 0.07 | 0.05 | 0.13 |
| +R-Edit | 0.85 | **0.60** | 0.48 | 0.64 | 0.27 | 0.12 | 0.04 | 0.14 | 0.30 | 0.09 | 0.05 | 0.15 | 0.26 | 0.06 | 0.04 | 0.12 |
| +WilKE | 0.81 | 0.59 | 0.44 | 0.61 | 0.45 | 0.27 | 0.19 | 0.30 | 0.28 | 0.10 | 0.10 | 0.16 | 0.18 | 0.02 | 0.07 | 0.09 |
| +PRUNE | 0.76 | 0.51 | 0.28 | 0.52 | 0.35 | 0.21 | 0.21 | 0.26 | 0.42 | 0.12 | 0.05 | 0.20 | 0.33 | 0.15 | 0.22 | 0.23 |
| +O-Edit | **0.99** | 0.51 | 0.73 | **0.74** | **0.68** | 0.41 | 0.37 | 0.49 | 0.45 | 0.18 | 0.26 | 0.30 | 0.37 | 0.20 | 0.19 | 0.25 |
| +O-Edit+ | 0.94 | 0.47 | 0.76 | 0.72 | 0.65 | **0.38** | **0.41** | 0.48 | 0.49 | 0.21 | 0.29 | **0.33** | 0.41 | 0.21 | 0.24 | **0.29** |
| MEMIT | 0.93 | 0.67 | 0.41 | 0.67 | 0.50 | 0.35 | 0.10 | 0.32 | 0.28 | 0.10 | 0.06 | 0.15 | 0.19 | 0.06 | 0.05 | 0.10 |
| +R-Edit | 0.93 | **0.64** | 0.48 | 0.68 | 0.76 | 0.39 | 0.16 | 0.44 | 0.32 | 0.17 | 0.06 | 0.18 | 0.28 | 0.13 | 0.06 | 0.16 |
| +WilKE | **0.95** | 0.70 | 0.50 | 0.72 | 0.73 | 0.51 | 0.26 | 0.50 | 0.26 | 0.16 | 0.06 | 0.16 | 0.30 | 0.14 | 0.04 | 0.16 |
| +PRUNE | 0.83 | 0.53 | 0.47 | 0.61 | 0.76 | 0.52 | 0.29 | 0.52 | 0.65 | 0.45 | 0.22 | 0.44 | 0.43 | 0.27 | 0.12 | 0.27 |
| +O-Edit | 0.93 | 0.55 | 0.65 | 0.71 | **0.86** | 0.53 | 0.45 | 0.61 | **0.72** | **0.47** | 0.34 | 0.51 | 0.51 | 0.33 | 0.18 | 0.34 |
| +O-Edit+ | 0.89 | 0.61 | **0.78** | **0.76** | 0.81 | **0.55** | **0.60** | **0.65** | 0.68 | 0.39 | **0.55** | **0.54** | **0.61** | **0.42** | **0.53** | **0.52** |
| +♠ O-Edit+ | **0.98** | **0.76** | **0.91** | **0.88** | **0.89** | **0.67** | **0.82** | **0.80** | **0.81** | **0.60** | **0.73** | **0.71** | **0.79** | **0.55** | **0.68** | **0.67** |
| **Llama3-8B** | | | | | | | | | | | | | | | | |
| FT | 0.24 | 0.09 | 0.11 | 0.14 | 0.07 | 0.02 | 0.01 | 0.03 | 0.04 | 0.01 | 0.01 | 0.02 | 0.02 | 0.01 | 0.00 | 0.01 |
| FT-EWC | 0.61 | 0.30 | 0.20 | 0.36 | 0.44 | 0.21 | 0.15 | 0.27 | 0.29 | 0.11 | 0.09 | 0.16 | 0.18 | 0.10 | 0.02 | 0.10 |
| MEND | 0.44 | 0.24 | 0.18 | 0.28 | 0.25 | 0.10 | 0.10 | 0.15 | 0.15 | 0.07 | 0.06 | 0.10 | 0.09 | 0.03 | 0.01 | 0.04 |
| ROME | 0.75 | 0.48 | 0.14 | 0.46 | 0.69 | 0.45 | 0.05 | 0.40 | 0.75 | 0.46 | 0.02 | 0.41 | 0.47 | 0.28 | 0.02 | 0.31 |
| +R-Edit | 0.70 | 0.38 | 0.27 | 0.45 | 0.65 | 0.41 | 0.06 | 0.37 | 0.54 | 0.34 | 0.03 | 0.30 | 0.50 | 0.31 | 0.02 | 0.28 |
| +WilKE | 0.77 | 0.44 | 0.33 | 0.51 | 0.55 | 0.42 | 0.03 | 0.33 | 0.66 | 0.45 | 0.02 | 0.38 | 0.71 | 0.49 | 0.02 | 0.41 |
| +PRUNE | **0.90** | 0.57 | 0.33 | 0.60 | 0.77 | 0.50 | 0.24 | 0.50 | 0.83 | 0.41 | 0.21 | 0.48 | 0.81 | 0.35 | **0.19** | 0.45 |
| +O-Edit | 0.88 | **0.63** | 0.35 | **0.62** | 0.77 | 0.47 | 0.22 | 0.49 | 0.84 | 0.47 | 0.13 | 0.48 | 0.83 | 0.31 | 0.09 | 0.41 |
| +O-Edit+ | 0.86 | 0.61 | **0.37** | 0.61 | **0.81** | **0.52** | 0.24 | **0.52** | **0.86** | **0.49** | 0.19 | **0.51** | **0.87** | **0.50** | 0.13 | **0.50** |
| MEMIT | 0.85 | 0.51 | 0.22 | 0.52 | 0.50 | 0.35 | 0.10 | 0.32 | 0.28 | 0.10 | 0.05 | 0.14 | 0.18 | 0.06 | 0.05 | 0.10 |
| +R-Edit | 0.92 | 0.63 | 0.48 | 0.68 | 0.57 | 0.39 | 0.15 | 0.37 | 0.34 | 0.17 | 0.06 | 0.19 | 0.27 | 0.13 | 0.05 | 0.15 |
| + WilKE | **0.95** | **0.68** | 0.50 | 0.71 | 0.71 | **0.56** | 0.25 | 0.51 | 0.30 | 0.16 | 0.08 | 0.18 | 0.30 | 0.14 | 0.05 | 0.16 |
| +PRUNE | 0.82 | 0.52 | 0.47 | 0.60 | 0.76 | 0.52 | 0.38 | 0.55 | 0.64 | 0.44 | **0.32** | 0.47 | 0.42 | 0.27 | 0.22 | 0.30 |
| +O-Edit | 0.93 | 0.55 | 0.64 | 0.71 | **0.86** | 0.53 | 0.44 | 0.61 | **0.72** | **0.47** | 0.33 | 0.51 | 0.55 | 0.40 | 0.27 | 0.41 |
| +O-Edit+ | 0.88 | 0.53 | **0.76** | **0.72** | 0.84 | 0.51 | **0.45** | **0.60** | 0.81 | 0.50 | 0.31 | **0.54** | **0.79** | **0.44** | **0.28** | **0.50** |
| +♠ O-Edit+ | **0.98** | 0.62 | **0.91** | **0.84** | **0.95** | 0.57 | **0.78** | **0.79** | **0.91** | 0.51 | **0.63** | **0.68** | **0.91** | 0.45 | **0.56** | **0.64** |

represents editing 100 pieces of knowledge at a time. Additionally, we considered the following methods: **R-Edit** (Gupta et al., 2024a), **WilKE** (Hu et al., 2024b), and **PRUNE** (Ma et al., 2024). See Appendix B.7 for methods details.

**Metrics.** Each edit example comprises an edit knowledge statement, consisting of an edit statement $\mathbf{x_e}$ and an edit target $\mathbf{y_e}$, its paraphrase sentences $\mathbf{x_{e'}}$ for testing generalization, and an unrelated knowledge statement $\mathbf{x_{loc}}$ for testing locality. For the editing dataset $\mathcal{D}_{edit} = \{(\mathbf{x_e}, \mathbf{y_e})\}$ with $T$ edits, we evaluate the final post-edit model $f_{\Theta_T}$ after the $T$-th edit example $(\mathbf{x_T}, \mathbf{y_T})$. We assess the reliability and generalization of the model editor using the metrics **Rel.** (Edit Success Rate (Zhang et al., 2024a)) and **Gen.** (Generalization Success Rate), while **Loc.** (Localization Success Rate) evaluates specificity, defined as the post-edit model's ability to maintain the output of the unrelated knowledge $\mathbf{x_{loc}}$. We report these metrics and their mean scores, which are formally defined as:

$$\text{Rel.} = \frac{1}{T}\sum_{t=1}^{T}\mathbb{1}(f_{\Theta_T}(\mathbf{x}_e^t) = \mathbf{y}_e^t), \text{Gen.} = \frac{1}{T}\sum_{t=1}^{T}\mathbb{1}(f_{\Theta_T}(\mathbf{x}_{e'}^t) = \mathbf{y}_e^t), \text{Loc.} = \frac{1}{T}\sum_{t=1}^{T}\mathbb{1}(f_{\Theta_T}(\mathbf{x}_{loc}^t) = f_{\Theta_0}(\mathbf{x}_{loc}^t)), \quad (11)$$

Here, $\mathbb{1}(\cdot)$ denotes the indicator function, which indicates that we only consider the top-1 token during inference. For RECENT and WIKICF, we have established additional evaluation metrics to assess the reasoning ability, subject alignment capability of editing methods, and more. For further details, please refer to Appendix B.10.

**Main Results.** The competitive performance of our methods is demonstrated in Tables 1. In the **COUNTERFACT** setting, with $T = 200$, models edited with MEMIT and ROME still perform effective edits. However, as the number of edits exceeds 500, their performance declines rapidly. After 1,500 edits on Mistral-7B, MEMIT's scores dropped to approximately 0.20 for **Rel.** and 0.05 for **Loc.**, indicating substantial forgetting of both edited and unrelated knowledge. Although improved methods like PRUNE and WilKE showed competitive performance at $T = 200$, they similarly failed to maintain a good balance across **Rel.**, **Gen.**, and **Loc.** at $T = 1500$. At $T = \{500, 1000, 1500\}$, O-Edit and O-Edit+ achieved the best results on both Mistral-7B and Llama3-8B. At $T = 1500$ with Mistral-7B, O-Edit+ improved by 0.16 and 0.42 in **Avg.** over ROME and MEMIT, respectively, and by 0.06 and 0.25 over PRUNE, our closest competitor. Overall, while performance across methods is similar for smaller numbers of edits, O-Edit+ significantly reduces forgetting as the number of edits increases, effectively preserving both edited and unrelated knowledge.

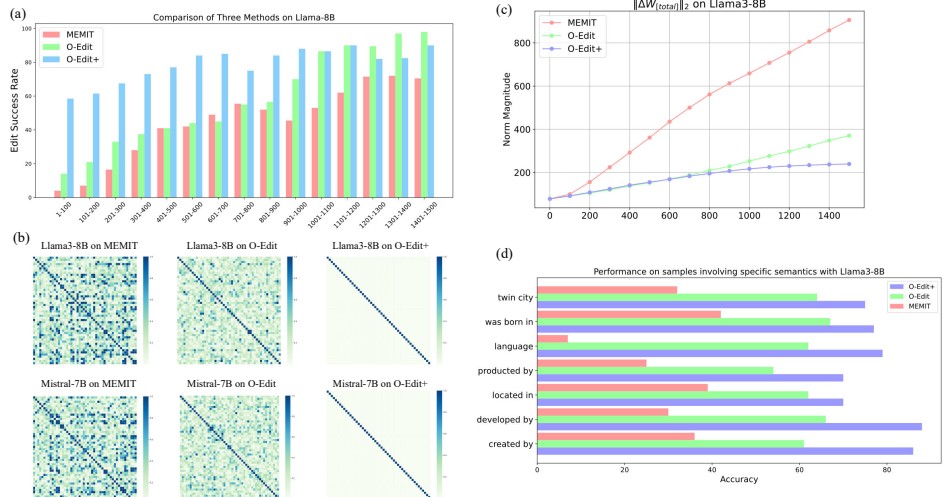

Figure 3: The further performance and impact of different editing methods include (a) editing success rates at different stages, (b) effects on update direction, (c) impact on matrix L2 norm, and (d) performance across different semantic relations.

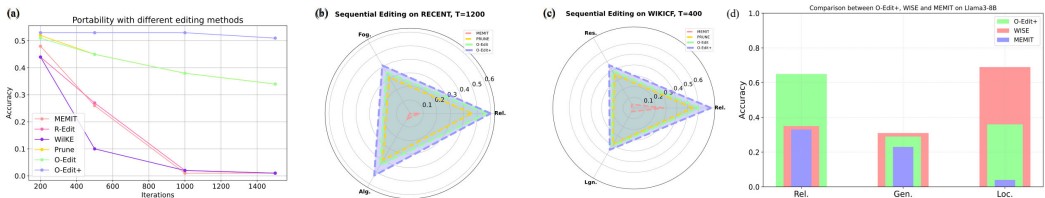

Figure 4: More performance metrics on the editing datasets: (a) Portability on the COUNTERFACT dataset, (b) performance on the RECENT dataset (Rel., Alg., Fog.), T=1200; (c) performance on the WIKICF dataset (Rel., Res., Lgn.), T=400. (d) Compared with the method of adding additional parameters, **WISE**, T=1500.

**The orthogonal editing method improves the success rate of edited knowledge across all stages.** We divided the editing process into 15 stages according to the sequence of edits, and evaluated the model after 1500 edits at each stage. As shown in Figure 3(a), the original method MEMIT exhibits a complete forgetting effect on the initial edits. In contrast, O-Edit shows significant improvements compared to MEMIT. Moreover, O-Edit performs best for edits between 1000 and 1500, demonstrating its ability to effectively retain recently edited knowledge. As for O-Edit+, it presents a balanced editing performance, excelling at updating both the initially edited knowledge and the recently edited knowledge.

**The orthogonal editing method altered the model's update direction.** We evaluated the orthogonality among each update matrix, $\Delta W_i$, by examining the cosine similarity between the corresponding update matrices after applying MEMIT, O-Edit, and O-Edit+. As illustrated in Figure 3(b), without any constraints, there is a significant overlap in the update directions, which may cause subsequent edits to influence the directions of prior edits. O-Edit mitigates this overlap by training an appropriate $v_*$, while O-Edit+ achieves complete orthogonality between each update direction through post-processing.

**The orthogonal editing method reduced the L2 norm of the matrix.** The L2 norm is considered by (Hu et al., 2024b) to be a key factor in limiting the effects of continuous editing. A larger L2 norm can lead to catastrophic forgetting. We visualized the change in the L2 norm of the matrices after multiple edits in Figure 3(c). For the unconstrained method, MEMIT exhibits a high growth trend in the L2 norm. In contrast, the orthogonal method reduces the growth trend of the matrix by constraining the model's update direction. We further discuss the impact of L2 norm on editing performance in Appendix B.12, revealing that not all methods of reducing the L2 norm improve the effectiveness of sequential editing.

**The orthogonal method performs better for editing any relation.** We selected seven representative semantic relations from COUNTERFACT for a cross-sectional comparison, as shown in Figure

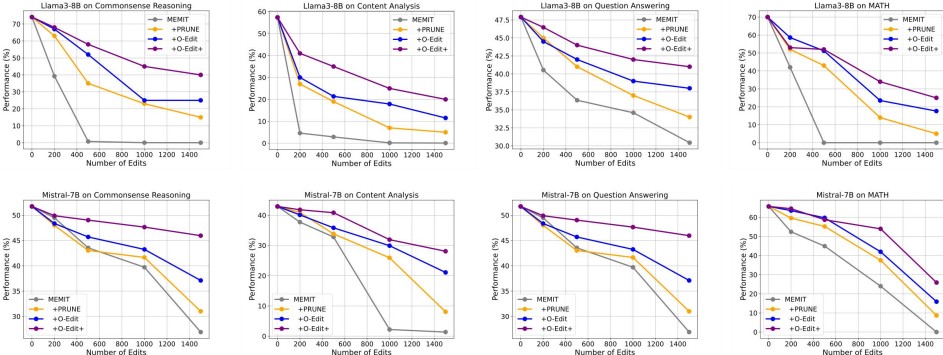

Figure 6: The downstream task performance (%) of models edited by four editing methods with Mistral-7B and Llama3-8B on the COUNTERFACT dataset.

3(d). The results indicate that both O-Edit and O-Edit+ exhibit higher editing accuracy for all relations, which aligns with the findings in Figure 3(a).

**The orthogonal editing method is applicable to the scaling laws of models.** To investigate this problem, we conducted tests on the GPT-2 series of models (Radford et al., 2019). To avoid inconsistencies in the semantic information extracted from the same layer across different models, we selected the middle layer of the model, $\lfloor \frac{\text{layers}}{2} \rfloor$, as the editing layer, which is referred to by (Meng et al., 2023a) as the place of knowledge storage. The experimental results are shown in Figure 5. As the size of the GPT models increases, the dimensions of each editing matrix also rise, leading to improved editing effects. The orthogonal editing method demonstrates varying degrees of enhancement across different models, with O-Edit+ achieving approximately double the performance across all models, indicating that the orthogonal editing methods are model-agnostic.

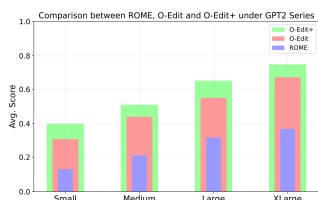

Figure 5: Comparsion under GPT2 Series, T=3000. Orthogonal editing is applicable to all model sizes.

**Further Results.** ❶ We followed the methodology in (Zhang et al., 2024a) to evaluate different editing methods across five additional metrics: Portability (Port.), Subject Aliasing (Alg.), Compositionality, Reasoning (Res.), Forgetfulness (Fog.), and Logical Generalization (Lgn.), as shown in Figure 4. For portability, O-Edit and O-Edit+ significantly outperform other methods, with O-Edit+ maintaining about 50% portability even after 1500 edits. Considering the dataset limitations and the complexity of the evaluation metrics, we chose to perform 1200 and 400 edits on the RECENT and WIKICF datasets, respectively. In both datasets, O-Edit and O-Edit+ consistently deliver the best editing performance, demonstrating their suitability for a wide range of editing scenarios. ❷ In addition, We further explored applying O-Edit and O-Edit+ for 3000 edits, with the results shown in Table 13. The original method completely forgets the previously edited and irrelevant knowledge, while O-Edit and O-Edit+ still maintain very good editing success rates. However, more edits lead to greater disruption of the original knowledge in the model, and the localization (Loc.) slightly decreases as the number of edits increases. For all extra experimental results, please refer to the Appendix B.13. ❸ We further compared the localized O-Edit+ method with the SOTA method, WISE (Wang et al., 2024b) that adds additional parameters when performing 1,500 edits. Following (Wang et al., 2024b), we conducted experiments using the ZsRE dataset and standardized the number of added or modified layers to 8, with results shown in Figure 4(d). When editing 1,500 times, O-Edit+ achieved significantly higher editing accuracy than WISE, while maintaining comparable generalization performance. Due to WISE's expanded parameter search space, it demonstrated better retention of unrelated knowledge, this comes at the cost of additional storage space and inference time. ❹ We have also discussed how to select the appropriate orthogonal space and the impact of orthogonality on editing performance in the Appendix B.9.

## 5.2 DOWNSTREAM TASKS EVALUATION

**Datasets.** To investigate the side effects of sequential model editing on the downstream task abilities of LLMs, we adopted four representative tasks with corresponding datasets for assessment: **Commonsense Reasoning** using the **SIQA** (Sap et al., 2019), **Content Analysis** on the **LAMBADA** (Paperno et al., 2016), **Question Answering** with the **CommonsenseQA** (Talmor et al., 2019), and **MATH** on the **GSM8K** (Cobbe et al., 2021).

**Main Results.** Figure 6 illustrates the downstream task performance of Mistral-7B and Llama3-8B after applying MEMIT and O-Edit+ in the **COUNTERFACT** setting. As shown by the gray line in Figure 6, MEMIT maintains performance at a certain level when the number of edits is small ($T \leq 200$). However, as the number of edits exceeds 1000, MEMIT's performance drastically declines, approaching zero (with results on CommonsenseQA resembling random guessing, both around 20%). In contrast, O-Edit and O-Edit+ effectively tackle this issue by implementing constraints that ensure orthogonality between the editing knowledge and the original model's implicit knowledge, significantly reducing interference. With O-Edit+ applied for 200 edits, downstream task performance remains close to that of the unedited model, effectively preserving accuracy across various tasks. Even after 1,500 edits, O-Edit+ remains to outperform both MEMIT and PRUNE, demonstrating its robustness in maintaining downstream task performance over extended sequences of edits. This highlights the effectiveness of O-Edit+ in minimizing interference between edits, allowing models to retain high performance even in heavily edited environments.

Nevertheless, as the number of edits increases, extensive knowledge editing inevitably leads to diminished model performance, a phenomenon described by (Wang et al., 2024b) as the "unbreakable triangle," which asserts that no method can achieve perfect editing without compromising other aspects of the model's performance. Despite this, O-Edit+ significantly mitigates this effect, offering superior performance retention compared to other editing methods such as MEMIT.

### 5.3 Further Analysis

**How do edits disturb model outputs?** We aim to study how each added piece of editing information affects the subsequent outputs of the model. Theoretically, if an editing method is effective, the output distribution of unrelated knowledge in the model should remain as consistent as possible with the pre-edit state when using this method. If the editing information is integrated into the subject's editing layer through newly created $(k_*, v_*)$ pairs, the information from $v_*$ will influence the hidden states of subsequent Relation Tokens (*"The SpaceX is located in"*) via the attention module and gradually propagate through decoding to impact the final model output. To investigate how this newly added information affects the hidden states of relation tokens, we conducted the following two sets of experiments:

We preserved the update matrix $\Delta W_i$ for each $i$-th edit from 1500 edits. Subsequently, we first measured the impact of adding a single $\Delta W_i$ on the final-layer hidden states of relation tokens for each edited piece of knowledge $i$. Then, we measured the impact of adding $\Delta W_{[\text{total}]} = \sum_{i=1}^{n} \Delta W_i$ on the final-layer hidden states of relation tokens. The results were dimensionally reduced using t-SNE, as shown in Figure 7 (a). It can be observed that the distribution difference between single and multiple edits in MEMIT is significant, indicating that multiple edits affect the model's final outputs. In contrast, the distributions for O-Edit+ show almost no difference, suggesting that the results of multiple edits do not affect the model's output distribution for each edited knowledge.

We also examined the distribution of relation tokens in the original model compared to the distribution after adding $\Delta W_{\text{unrelated}} = \Delta W_{[\text{total}]} - \Delta W_j$. Theoretically, $\Delta W_{\text{unrelated}}$ should carry no meaningful information for the edited knowledge $j$, and we expect the distribution after adding $\Delta W_{\text{unrelated}}$ to remain consistent with $W_{\text{original}}$. The experimental results are shown in Figure 7 (b). It can be seen that using O-Edit+ with $\Delta W_{\text{unrelated}}$ has almost no effect on the edited knowledge $j$, while MEMIT causes a shift in the distribution.

**How do edits disturb each other?** To investigate the extent of interdependencies among knowledge updates during the sequential editing process, we preserved the update matrix $\Delta W_i$ for each $i$-th edit. Upon completion of the sequential editing, the model's cumulative update matrix is computed as $\Delta W_{[\text{total}]} = \sum_{i=1}^{n} \Delta W_i$. For the $j$-th edit, we compute $\Delta W_{\text{unrelated}} = \Delta W_{[\text{total}]} - \Delta W_j$, which excludes the update matrix $\Delta W_j$ corresponding to $k_j$. According to Hu et al. (2024a), under ideal sequential editing, the knowledge vector $k_j$ used during the $j$-th edit should not activate any unrelated $\Delta W_{\neq j}$ (i.e., any update matrix other than $\Delta W_j$), meaning $\|\Delta W_{\text{unrelated}} \cdot k_j\|_2 = 0$. We calculate the activation score (AS) for each edit as $\|\Delta W_{\text{unrelated}} \cdot k_j\|_2$. As illustrated in Figure 7 (c), after 1,500 edits, the original method exhibited high activation scores (AS), with some values reaching approximately 2.5 and others exceeding 10. This indicates that in the original method, any unrelated $\Delta W_{\neq j}$ (i.e., any update matrix other than $\Delta W_j$) could significantly activate $k_j$, leading to a substantial deviation from the ideal state $v_*$ and resulting in the failure of MEMIT in sequential

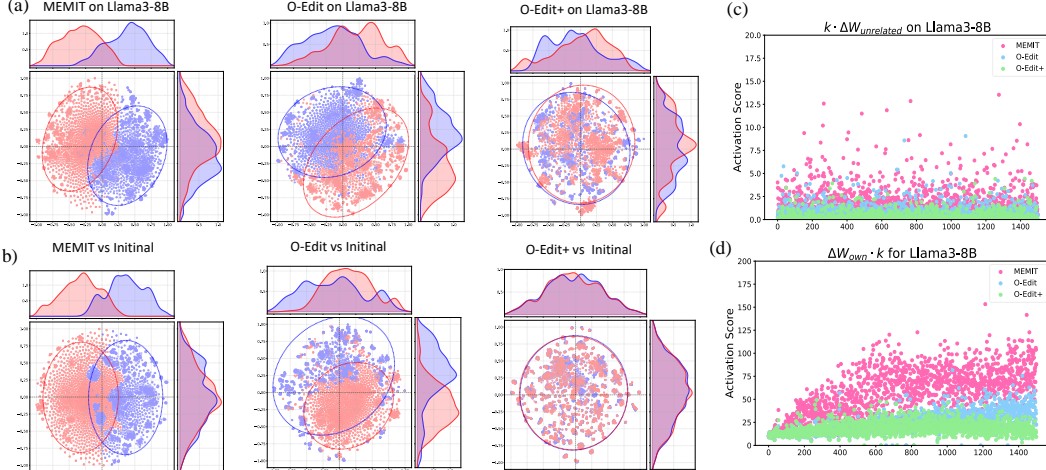

Figure 7: The impact of different editing methods on the model's operation mechanism. (a) The distribution of hidden representations of different editing methods in post-edited LLMs after dimensionality reduction. (b) The distribution of hidden representations with or without editing methods. (c) The activation score caused by unrelated parameters. (d) The activation score caused by a single update parameter.

editing. In contrast, both O-Edit and O-Edit+ consistently achieved activation values below 2.5 for nearly all edits, with some values approaching zero. In Appendix B.12, we analyze the reasons for this phenomenon from a mathematical derivation perspective, highlighting that the key lies in the orthogonality of the column subspaces of each update matrix.

We aim to further understand the interaction between the $j$-th $k_j$ and $\Delta W_j$. We calculate the activation score (AS) for each edit as $\|\Delta W_j \cdot k_j\|_2$ ($\|\Delta W_{\text{own}} \cdot k_j\|_2$), as illustrated in Figure 7 (d). After 1500 edits, the activation values in MEMIT gradually increase with the number of edits due to the significant activation value $\|\Delta W_{<j} \cdot k_j\|_2$ ($\|\Delta W_{\text{unrelated}} \cdot k_j\|_2$). This phenomenon occurs because completing an edit requires a larger activation value to counteract the influence of previous edits, resulting in a vicious cycle and ultimately poor sequential editing performance. In contrast, the activation values for $\|\Delta W_{<j} \cdot k_j\|_2$ ($\|\Delta W_{\text{unrelated}} \cdot k_j\|_2$) in O-Edit and O-Edit+ remain consistently low, indicating that a large activation value for $\|\Delta W_{\text{own}} \cdot k_j\|_2$ is not necessary to complete a new edit. Consequently, although the activation values are small, O-Edit and O-Edit+ allow for a greater number of effective edits.

## 6 LIMITATIONS

While O-Edit and O-Edit+ demonstrate robust sequential editing performance, several limitations persist. Due to computational constraints, we restricted our experiments to Mistral-7B and Llama3-8B, leaving the scalability of our methods on larger models untested. Additionally, constructing orthogonality between edits adds computational overhead, which may prolong editing times. However, O-Edit and O-Edit+ require maintaining only two additional matrices, making them both model-agnostic and compatible with other sequential editing techniques. Furthermore, we did not evaluate O-Edit and O-Edit+ against other editing methods, such as fine-tuning (FT), as these approaches tend to falter after only a few sequential edits, whereas ROME and MEMIT can support more extensive editing sequences. Despite these challenges, we believe our methods hold promising potential, particularly in the early stages of research on sequential model editing.

## 7 CONCLUSION

In this paper, we present two innovative methods—O-Edit and O-Edit+ that leverage orthogonal subspace editing for sequential knowledge editing in language models. These methods effectively mitigate catastrophic forgetting of both edited and existing knowledge by incrementally applying edits in orthogonal subspaces. Our methods distinguish themselves through their attention to data privacy, efficient parameter utilization, and strong generalization capabilities for downstream tasks. Comprehensive empirical evaluations indicate that O-Edit and O-Edit+ significantly outperform existing methods, establishing them as promising avenues for future advancements in sequential knowledge editing.

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

# A  RELATED WORK

## A.1  KNOWLEDGE EDITING

From the perspective of whether model parameters are modified, (Yao et al., 2023) categorized knowledge editing methods into two major classes: preserving the model's parameters and modifying the model's parameters. This paper primarily focuses on the latter. On one hand, meta-learning has been used to predict parameter updates for networks, typically employing a hypernetwork to edit language models. (Cao et al., 2021) used a bidirectional LSTM to predict weight updates for editing, while (Mitchell et al., 2022a) utilized low-rank decomposition of gradients to fine-tune language models, known as MEND, and (Tan et al., 2024) extended single-step edits to batch edits using a least squares method based on MEND. On the other hand, (Meng et al., 2023a; Dai et al., 2022) employed a causal probe to localize knowledge within the intermediate layers of the model, demonstrating that editing in the MLP of the middle layers yields the best results. (Dai et al., 2022) performed knowledge editing by modifying the activation values of specific neurons. (Meng et al., 2023a) used a constrained least squares method to precisely solve for the parameter updates required for editing and extended this approach to batch editing (Meng et al., 2023b).

## A.2  SEQUENTIAL EDITING

Some studies have extended knowledge editing methods to sequential editing. From the perspective of modifying model parameters, (Ma et al., 2024) theoretically analyzed that the bottleneck limiting sequential editing in models lies in the condition number of matrices, and they attempted to support sequential editing by controlling the growth of the matrix condition number. (Hu et al., 2024b) attributed the decline in performance during sequential editing to pattern mismatch, where different layers detect different patterns, making a single layer incapable of accommodating all the edited knowledge. Thus, they selected the optimal layer from multiple layers for editing. Additionally, (Hu et al., 2024a) explored the root causes of failures in sequential editing, deriving a closed-form solution from linear associative memory. They posited that lossless sequential editing can only be achieved when the edited knowledge is completely orthogonal. From the perspective of adding additional parameters while freezing model parameters, SERAC (Mitchell et al., 2022b) stores edits in memory. When an input is received, a classifier checks whether it corresponds to any cached edits. If a match is found, a counterfactual model uses the input and relevant edits to predict outputs. GRACE (Hartvigsen et al., 2023) uses semantic similarity in the model's latent space by adding an offline key-value adapter at the selected layers, applying edits only to inputs that are similar to the keys cached in the encoding. WISE (Wang et al., 2024b) uses a dual-parameter storage scheme, where the main memory is used for pre-trained knowledge and the side memory is designated for edited knowledge. By incorporating a knowledge sharding mechanism, it allows for editing knowledge in different parameter subspaces and merges them into the shared side memory without causing conflicts. In this paper, we consider the scenario of directly updating model parameters.

## A.3  CONTINUAL LEARNING

The orthogonal concept presented in this paper is inspired by continual learning. Existing continual learning methods typically update all tasks within a shared vector space (Ke & Liu, 2023), which directly affects the model's hidden layer outputs (Wang et al., 2024a). Some studies (Farajtabar et al., 2019; Saha et al., 2021) have proposed a promising approach to address this issue by performing gradient descent optimization in directions orthogonal to the gradient subspaces of past tasks, effectively mitigating catastrophic forgetting. GPM (Saha et al., 2021) divides the gradient space into two key areas: the "Core Gradient Space" (CGS) and the "Residual Gradient Space" (RGS). By learning in the orthogonal directions of the CGS related to previous task inputs, it ensures minimal interference with past tasks. Based on GPM, TRGP (Lin et al., 2022) introduces a "trust region" concept to select old tasks relevant to new ones, reusing their frozen weights through scaled weight projections. By optimizing the scaling matrix and updating the model along orthogonal directions to the old tasks' subspace, TRGP effectively facilitates knowledge transfer without forgetting. O-LoRA (Wang et al., 2023b) suggests that parameter information updated through low rank can be approximately equivalent to gradient information, which expands the application scenarios of continual learning and enables effective learning even in scenarios where gradient information cannot be obtained.

# B ALGORITHM

## B.1 ORTHOGONAL GRADIENT DESCENT FOR CONTINUAL LEARNING

Consider a continual learning setting where tasks $\{T_1, T_2, T_3, \ldots\}$ are learned sequentially without access to previous task data. Suppose that the model has been trained on $T_A$ in the usual way until convergence to a update parameter $w_A^*$. To mitigate the impact on $T_A$ while training on the next task $T_B$, Farajtabar et al. (2019) propose to "orthogonalize" it in a way that the new update direction $\tilde{g}$ on $T_B$ satisfies:

$$\tilde{g} \perp \nabla f(x; w_A^*), \quad \forall x \in T_A. \tag{12}$$

One can compute and store $\nabla f(x; w)$ for all $x \in T_A$ when training on $T_A$ is done. In a continual learning scenario involving multiple tasks, the direction of gradient updates is determined by:

$$\tilde{g} = g - \sum_{i=1}^{n_A} \text{proj}_{\mathbf{g}_i}(g) = g - \sum_{i=1}^{n_A} \langle g, \mathbf{g}_i \rangle \, \mathbf{g}_i \tag{13}$$

The new direction $-\tilde{g}$ is still a descent direction for $T_B$, meaning that there exists $\epsilon > 0$ such that for any learning rate $0 < \eta < \epsilon$, taking the step $-\eta\tilde{g}$ reduces the loss.

## B.2 SINGULAR VALUE DECOMPOSITION AND RANK-$r$ APPROXIMATION

Singular Value Decomposition (SVD) decomposes any matrix $W \in \mathbb{R}^{m \times n}$ into three matrices: $W = U\Sigma V^T$, where $U \in \mathbb{R}^{m \times m}$ and $V \in \mathbb{R}^{n \times n}$ are orthogonal matrices, and $\Sigma$ is a diagonal matrix containing the singular values $\sigma_i$ of $W$, ordered in descending magnitude. SVD is instrumental in solving the rank-$r$ approximation problem, where the goal is to find a matrix $\tilde{W}$ that minimizes $\|\tilde{W} - W\|_2$ subject to $\text{rank}(\tilde{W}) \leq r$. According to the Eckart–Young–Mirsky theorem (Eckart & Young, 1936), the optimal rank-$r$ approximation $\tilde{W}$ is given by $\tilde{W} = \sum_{i=1}^{r} \sigma_i u_i v_i^T$, obtained by truncating the SVD of $W$ to retain the top $r$ singular values and their corresponding singular vectors, where $r \leq \min\{m, n\}$.

## B.3 ROME

In their study, (Meng et al., 2023a) employed causal mediation analysis to identify that feed-forward neural networks (FFNs) play a crucial role in retaining factual knowledge. The FFN is decomposed into two matrices, represented as follows:

$$FFN^l(x) = W_{proj}^l \cdot \sigma(W_{fc}^l \cdot \gamma(a^l + h^{l-1})) \tag{14}$$

Here, $a^l \in \mathbb{R}^d$ represents the output of the attention module at the $l$-th layer, and $h^{l-1} \in \mathbb{R}^d$ denotes the output of the previous layer. The matrices $W_{fc}^l \in \mathbb{R}^{d_m \times d}$ and $W_{proj}^l \in \mathbb{R}^{d \times d_m}$ serve as the parameter matrices for the FFN at the $l$-th layer. Here, $d_m$ is the dimension of the intermediate hidden state, $\sigma$ denotes the activation function, and normalizing nonlinearity $\gamma$.

Building on the key-value memory theory introduced in (Geva et al., 2021; 2022), the matrix $W_{fc}^l$ is responsible for identifying input patterns, which leads to the generation of the key vector $k \in \mathbb{R}_m^d$. In contrast, $W_{proj}^l$ retrieves the corresponding value vector $v \in \mathbb{R}^d$. This establishes $W_{proj}^l$ as a linear key-value memory system, where the set of key vectors $K = \{k_1, k_2, \ldots\}$ is associated with the corresponding set of value vectors $V = \{v_1, v_2, \ldots\}$. The relationship between the keys and values can be succinctly expressed as $WK = V$, thereby completing the transformation process.

Meng et al. (2023a) propose **ROME**, in which new knowledge is represented as a key-value pair $(k_*, v_*)$ and is integrated into the model by addressing the following constrained least squares problem:

$$\min \|\widetilde{W}K - V\|_2 \quad \text{subject to} \quad \widetilde{W}k_* = v_*, \quad \text{with} \quad \widetilde{W} = W + \Lambda(C^{-1}k_*)^T. \tag{15}$$

Here, $\Delta W = \Lambda(C^{-1}k_*)^T$, $k_*$ represents the query associated with the knowledge to be edited, such as $x =$ *"The president of the US is"*, where $k_*$ corresponds to the hidden state of the last token (index $i$) of the subject (e.g., *"US"*). The key vector $k_*$ is defined as:

$$k_* = \frac{1}{N}\sum_{j=1}^{N} k(s_j + x), \quad \text{where } k(x) = \sigma\left(W_{fc}^l\gamma\left(a_{[x],i}^l + h_{[x],i}^{l-1}\right)\right), \tag{16}$$

with $s_j$ representing prefix texts for robustness. The value vector $v_*$ denotes the edited knowledge result, for instance, *"Harris"* or *"Trump"*, computed as $v_* = \arg\min_v \mathcal{L}(v)$, where $\mathcal{L}(v)$ is given by:

$$\mathcal{L}(v) = \frac{1}{N}\sum_{j=1}^{N} -\log P_{(v=v_*)}[o^*|p_j + x] + D_{KL}\left(P_{G(v=v_*)}[x|p'] \parallel P_G[x|p']\right). \tag{17}$$

The first term serves to update the knowledge, while the second term preserves the essence of the subject. The objective is to modify the model's response to the knowledge query, yielding an output $o^*$ (e.g., *"Harris"* or *"Trump"*). Additionally, $C = KK^T$ is a pre-computed constant that estimates the uncentered covariance of $k$, and $\Lambda = (v_* - Wk_*)/(C^{-1}k_*)^T k_*$ represents the residual error of the new key-value pair. Further details can be found in (Meng et al., 2023a).

To manage editing intensity, (Meng et al., 2023b) introduced **MEMIT**, which computes matrix updates by solving:

$$\widetilde{W} = W + Rk_*^T(C + k_*k_*^T)^{-1}, \tag{18}$$

where $\Delta W = Rk_*^T(C + k_*k_*^T)^{-1}$, $C = \lambda \cdot KK^T$, and $R = v_* - Wk_* \in \mathbb{R}^d$ is a column vector. The parameter $\lambda$ allows for adjusting the balance between new edits and the original knowledge. It is noteworthy that in both ROME and MEMIT, **only $v_*$ is derived through the training process**, and this operation will be optimized in subsequent steps. For additional implementation details regarding MEMIT, please refer to Appendix B.4.

## B.4 MEMIT

In this paper, we consider the scenario of editing one piece of knowledge at a time. Similar to ROME, MEMIT views $W_{\text{proj}}^l$ as a linear key-value memory for a set of vector keys $K = \{k_1, k_2, \ldots\}$ and corresponding vector values $V = \{v_1, v_2, \ldots\}$ by solving $WK = V$. It attempts to insert a new key-value pair $(k_*, v_*)$ into the model by solving the following constrained least squares problem:

$$\widetilde{W} = \underset{\hat{W}}{\arg\min}\left(\left\|\hat{W}K - V\right\|_2 + \left\|\hat{W}k_* - v_*\right\|_2\right). \tag{19}$$

MEMIT solves Eqn. 19 by applying the normal equation, which is expressed in block form:

$$\widetilde{W}\begin{bmatrix} K & k_* \end{bmatrix}\begin{bmatrix} K^T \\ k_*^T \end{bmatrix} = \begin{bmatrix} V & v_* \end{bmatrix}\begin{bmatrix} K^T \\ k_*^T \end{bmatrix}, \tag{10}$$

which expands to:

$$(W + \Delta)\left(KK^T + k_*k_*^T\right) = VK^T + v_*k_*^T, \tag{11}$$

$$WKK^T + Wk_*k_*^T + \Delta K^T K + \Delta k_*^T k_* = VK^T + v_*k_*^T. \tag{12}$$

Under the condition $WK = V$, we can simplify to:

$$\Delta(KK^T + k_*k_*^T) = v_*k_*^T - Wk_*k_*^T, \tag{20}$$

yielding:

$$\Delta = (v_* - Wk_*)k_*^T(KK^T + k_*k_*^T)^{-1}. \tag{21}$$

Thus, the final update rule is:

$$\widetilde{W} = W + (v_* - Wk_*)k_*^T(KK^T + k_*k_*^T)^{-1}. \tag{22}$$

Here, $(v_* - Wk_*) \in \mathbb{R}^d$ is a column vector, and $k_*^T(KK^T + k_*k_*^T)^{-1} \in \mathbb{R}^{d_m}$ is a row vector. By adjusting the hyperparameter $\lambda$, MEMIT balances the preservation of existing knowledge and the incorporation of new edits. Consequently, the updated equation is expressed as follows:

$$\widetilde{W} = W + (v_* - Wk_*)k_*^T(\lambda KK^T + k_*k_*^T)^{-1}. \tag{23}$$

Like ROME, $KK^T$ is pre-cached by estimating the uncentered covariance of $k$ from a sample of Wikipedia text. The rank of the update matrix $\Delta W = (v_* - Wk_*)k_*^T(\lambda KK^T + k_*k_*^T)^{-1}$ obtained through ROME and MEMIT is 1.

In fact, MEMIT is a scalable extension of ROME. By increasing $\lambda$, MEMIT effectively enhances the retention of existing knowledge while also allowing for new updates. However, the restrictive conditions imposed by ROME, which require $k_*\widetilde{W} \equiv v_*$ as seen in Eqn. 15, can be overly stringent and may lead to greater disruption of existing knowledge within the model.

### B.5 O-EDIT AND O-EDIT+

**We will provide a detailed explanation of the calculation formula for O-Edit.** To explain how to compute Eqn. 5, we first analyze the properties of the update matrices for each piece of knowledge. Based on the matrix property $\text{rank}(AB) \leq \min(\text{rank}(A), \text{rank}(B))$, the ranks of $\Lambda(C^{-1}k_*)^T$ in Eqn. 15 and $Rk_*^T(C + k_*k_*^T)^{-1}$ in Eqn. 18 are both 1. In the $i$-th edit, the rank of the cached $\Delta W_{[\text{total}]} \in \mathbb{R}^{d \times d_m}$ is at most $i$, with equality when each $k_*$ is linearly independent. After several edits, $\text{rank}(\Delta W_{[\text{total}]}) = 1 \times \text{iteration}$, but as updates increase, $\text{rank}(\Delta W_{[\text{total}]})$ may fall below the iteration count. Therefore, $r$ is always equal to $\text{rank}(\Delta W_{[\text{total}]})$, and $\Delta W_r$ is $\Delta W_{[\text{total}]}$ itself.

During the computation process, we observe that $\Delta W_r^T = U_{\Delta W_r}\Sigma_{\Delta W_r}V_{\Delta W_r}^T$, where $U_{\Delta W_r} \in \mathbb{R}^{d_m \times r}$, $V_{\Delta W_r} \in \mathbb{R}^{d \times r}$, and $\Sigma_{\Delta W_r}$ is a diagonal matrix. Eqn. 2 can be rewritten as:

$$U_{\Delta W_r}\Sigma_{\Delta W_r}V_{\Delta W_r}^T \cdot \Delta W_{[2]} = \mathbf{0}. \tag{24}$$

We only need to ensure that $v_* - Wk_*$ is orthogonal to $V_r$. Therefore, Eqn. 5 can be rewritten as:

$$f_1 = \frac{1}{r}\sum_{i=0}^{r}\text{sim}(V_{\Delta W_r}[i], (v_* - Wk_*)). \tag{25}$$

The key reason for using cosine similarity instead of $V_{\Delta W_r}^T \cdot (v_* - Wk_*)$ is that the latter may lead to trivial solutions, i.e., $v_* - Wk_* = \mathbf{0}$, while cosine similarity considers angular information. In fact, merely reducing the norm of $v_* - Wk_*$ does not effectively enhance the effectiveness of sequential editing. The success of O-Edit and O-Edit+ lies in identifying the correct update direction during the sequential editing process. For further details, see **Further Analysis** 5.3.

Furthermore, when calculating $\nabla G$, we utilized a large amount of natural text, resulting in $\nabla G$ being a high-rank matrix, which is distinct from $\Delta W_{[\text{total}]}$. We dynamically adjust $q$ to select the core gradient subspace (CGS) of $\nabla G$, defined as $\nabla G_q^T = U_{\nabla G_q}\Sigma_{\nabla G_q}V_{\nabla G_q}^T$. The purpose of this adjustment is to counteract the cumulative impact of edited knowledge on the implicit knowledge within the model as the number of edits increases. We adjust $q$ to increase linearly with the number of edits. In practice, we compute Eqn. 7 by removing the projection of $V_{\nabla G_q}$ onto $V_{\Delta W_r}$:

$$V_{\nabla G_q} = V_{\nabla G_q} - V_{\Delta W_r}V_{\Delta W_r}^T V_{\nabla G_q}. \tag{26}$$

Finally, we compute Eqn. 9 as follows:

$$f_2 = \frac{1}{q}\sum_{i=0}^{q}\text{sim}(V_{\nabla G_q}[i], (v_* - Wk_*)). \tag{27}$$

**Next, we will provide a detailed explanation of the calculation formula for O-Edit+.** To ensure that the column subspaces of $\Delta W_r$ and $\Delta W_{[2]}$ are orthogonal, it is sufficient to ensure that the projection of $\Delta W_{[2]}$ onto the standard orthogonal basis of the column space of $\Delta W_r$ is zero. Similar

to O-Edit, $\Delta W_r$ is $\Delta W_{[\text{total}]}$, and $\nabla G_q$ is a high-rank matrix. Eqn. 10 can be rewritten as:

$$\Delta W_{[2]} = \Delta W_{[2]} - V_{\Delta W_r} V_{\Delta W_r}^T \Delta W_{[2]},$$
$$V_{\nabla G_q} = V_{\nabla G_q} - V_{\Delta W_r} V_{\Delta W_r}^T V_{\nabla G_q}, \tag{28}$$
$$\Delta W_{[2]} = \Delta W_{[2]} - V_{\nabla G_q} V_{\nabla G_q}^T \Delta W_{[2]}.$$

O-Edit and O-Edit+ are adaptations of ROME and MEMIT for sequential editing, and all experimental settings are consistent with those of ROME and MEMIT. Readers can refer to Algorithm 1 and Algorithm 2 for their pseudo-code.

---

**Algorithm 1** Algorithm for Sequential Editing with O-Edit

---

**Require:** $\mathcal{D}_{\text{edit}} = \{(\mathcal{X}_e, \mathcal{Y}_e) \mid (x_1, y_1), \ldots, (x_T, y_T)\}$, original weight $W$, hyperparamter $r, q, \lambda_1$, $\lambda_2, \lambda_3$, gradient information $\nabla G$.
**Ensure:** The optimal parameter $\widetilde{W}$
1: **for** Iteration $\in T$ **do**
2:     **if** Iteration $= 1$ **then**
3:         $q \leftarrow \lambda_3 \times 1, r \leftarrow 0$
4:         $\nabla G_q^T = U_{\nabla G_q} \Sigma_{\nabla G_q} V_{\nabla G_q}^T \leftarrow \|\nabla G_q - \nabla G\|_2$, subject to $\text{rank}(\nabla G_q) = q$ // Obtain by calculating the SVD decomposition of $\nabla G$
5:         Compute $k_* = \frac{1}{N} \sum_{j=1}^N k(s_j + x)$ (Eqn. 16)
6:         Compute $v_*$ by optimizing $\mathcal{L}(v) + 0 \cdot f_1(\Delta W_r; v) + \lambda_2 f_2(\nabla G_q; v)$ (Eqn.8) // Eqn. 25, 27 for compute $f_1$ and $f_2$.
7:         $\Delta W_{[1]} \leftarrow \Lambda(C^{-1}k_*)^T$ for ROME (Eqn.15) // $\Delta W_1 \leftarrow Rk_*^T(C + k_* k_*^T)^{-1}$ for MEMIT (Eqn. 18)
8:         $\widetilde{W} \leftarrow W + \Delta W_{[1]}$ // Update original weight $W$ to $\widetilde{W}$
9:         **Initialize** $\Delta W_{[\text{total}]} \leftarrow \Delta W_{[1]}$
10:     **else**
11:         $q \leftarrow \lambda_3 \times \text{Iteration}, r \leftarrow \min(1 \times \text{Iteration - 1}, \text{rank}(\Delta W_{[\text{total}]}))$
12:         $\nabla G_q^T = U_{\nabla G_q} \Sigma_{\nabla G_q} V_{\nabla G_q}^T \leftarrow \|\nabla G_q - \nabla G\|_2$, subject to $\text{rank}(\nabla G_q) = q$
13:         $\Delta W_r^T = U_{\Delta W_r} \Sigma_{\Delta W_r} V_{\Delta W_r}^T \leftarrow \|\Delta W_r - \Delta W_{[\text{total}]}\|_2$, subject to $\text{rank}(\Delta W_r) = r$ // Actually, $\Delta W_r = \Delta W_{[\text{total}]}$
14:         $\nabla G_q = \nabla G_q - \Delta W_r(\Delta W_r^T \Delta W_r)^{-1} \Delta W_r^T \nabla G_q$, // Avoid knowledge conflicts, compute by Eqn.26
15:         Compute $k_* = \frac{1}{N} \sum_{j=1}^N k(s_j + x)$ (Eqn. 16)
16:         Compute $v_*$ by optimizing $\mathcal{L}(v) + \lambda_1 f_1(\Delta W_r; v) + \lambda_2 f_2(\nabla G_q; v)$ (Eqn.8) // Eqn. 25, 27 for compute $f_1$ and $f_2$.
17:         $\Delta W_{[\text{Iteration}]} \leftarrow \Lambda(C^{-1}k_*)^T$ for ROME (Eqn.15) // $\Delta W_{[\text{Iteration}]} \leftarrow Rk_*^T(C + k_* k_*^T)^{-1}$ for MEMIT (Eqn. 18)
18:         $\widetilde{W} \leftarrow \widetilde{W} + \Delta W_{[\text{Iteration}]}$ // Iterative update of the model weights
19:         $\Delta W_{[\text{total}]} + = \Delta W_{[\text{Iteration}]}$ // Update the cache of $\Delta W_{[\text{total}]}$
20:     **end if**
21: **end for**
22: **return** update weight $\widetilde{W}$

---

### B.6 HOW TO CHOOSE AN APPROPRIATE $\nabla G_q$

The core of our method lies in capturing the update direction of implicit knowledge within the model. Theoretically, if we view the model as a knowledge base (Petroni et al., 2019), the update direction should align with the gradient direction in which the model continues to learn from this knowledge. Thus, selecting the appropriate knowledge base is crucial for determining the model's update gradient. We explored the following methods:

- We selected 100,000 pieces of unrelated knowledge from COUNTERFACT, which are outside the experimental test samples. This set, referred to as "locality_prompt" in Figure 9, serves as the expected gradient direction.

---

**Algorithm 2** Algorithm for Sequential Editing with O-Edit+

---

**Require:** $\mathcal{D}_{\text{edit}} = \{(\mathcal{X}_e, \mathcal{Y}_e) \mid (x_1, y_1), \ldots, (x_T, y_T)\}$, original weight $W$, hyperparamter $r$, $q$, $\lambda_3$, gradient information $\nabla G$.

**Ensure:** The optimal parameter $\widetilde{W}$

1: **for** Iteration $\in T$ **do**
2:     **if** Iteration $= 1$ **then**
3:         $q \leftarrow \lambda_3 \times 1, r \leftarrow 0$
4:         $\nabla G_q^T = U_{\nabla G_q} \Sigma_{\nabla G_q} V_{\nabla G_q}^T \leftarrow \|\nabla G_q - \nabla G\|_2$, subject to $\text{rank}(\nabla G_q) = q$ // Obtain by calculating the SVD decomposition of $\nabla G$
5:         Compute $k_* = \frac{1}{N} \sum_{j=1}^{N} k(s_j + x)$ (Eqn. 16)
6:         Compute $v_*$ by optimizing $\mathcal{L}(v)$ (Eqn.17)
7:         $\Delta W_1 \leftarrow \Lambda(C^{-1}k_*)^T$ for ROME (Eqn.15) // $\Delta W_{[1]} \leftarrow Rk_*^T(C + k_*k_*^T)^{-1}$ for MEMIT (Eqn. 18)
8:         $\Delta W_{[1]} = \Delta W_{[1]} - V_{\nabla G_q} V_{\nabla G_q}^T \Delta W_{[1]}$. (Eqn. 28)// Orthogonal post-processing
9:         $\widetilde{W} \leftarrow W + \Delta W_{[1]}$ // Update original weight $W$ to $\widetilde{W}$
10:        **Initialize** $\Delta W_{[\text{total}]} \leftarrow \Delta W_{[1]}$
11:     **else**
12:         $q \leftarrow \lambda_3 \times \text{Iteration}, r \leftarrow \text{Iteration} - 1$
13:         $\nabla G_q^T = U_{\nabla G_q} \Sigma_{\nabla G_q} V_{\nabla G_q}^T \leftarrow \|\nabla G_q - \nabla G\|_2$, subject to $\text{rank}(\nabla G_q) = q$
14:         $\Delta W_r^T = U_{\Delta W_r} \Sigma_{\Delta W_r} V_{\Delta W_r}^T \leftarrow \|\Delta W_r - \Delta W_{[\text{total}]}\|_2$, subject to $\text{rank}(\Delta W_r) = r$ // Actually, $\Delta W_r = \Delta W_{[\text{total}]}$
15:         Compute $k_* = \frac{1}{N} \sum_{j=1}^{N} k(s_j + x)$ (Eqn. 16)
16:         Compute $v_*$ by optimizing $\mathcal{L}(v)$ (Eqn.17)
17:         $\Delta W_{[\text{Iteration}]} \leftarrow \Lambda(C^{-1}k_*)^T$ for ROME (Eqn.15) // $\Delta W_{\text{Iteration}} \leftarrow Rk_*^T(C + k_*k_*^T)^{-1}$ for MEMIT (Eqn. 18)
18:         $\Delta W_{[\text{Iteration}]} = \Delta W_{[\text{Iteration}]} - V_{\Delta W_r} V_{\Delta W_r}^T \Delta W_{[\text{Iteration}]}$
           $V_{\nabla G_q} = V_{\nabla G_q} - V_{\Delta W_r} V_{\Delta W_r}^T V_{\nabla G_q}$, (Eqn. 28) // Orthogonal post-processing
           $\Delta W_{[\text{Iteration}]} = \Delta W_{[\text{Iteration}]} - V_{\nabla G_q} V_{\nabla G_q}^T \Delta W_{[\text{Iteration}]}$
19:         $\widetilde{W} \leftarrow \widetilde{W} + \Delta W_{[\text{Iteration}]}$ // Iterative update of the model weights
20:         $\Delta W_{[\text{total}]}+ = \Delta W_{[\text{Iteration}]}$ // Update the cache of $\Delta W_{[\text{total}]}$
21:     **end if**
22: **end for**
23: **return** update weight $\widetilde{W}$

---

- We utilized the knowledge employed by (Meng et al., 2023a), which successfully identified how knowledge is stored within the model.
- For comparison, we randomly generated 100,000 text samples using ASCII codes.
- We also used Wikipedia as a knowledge source, as it is commonly chosen for pre-training in large language models (LLMs).

The experimental results are presented in Appendix Table 2. We maintained consistency in the parameters related to $\Delta W_{[\text{total}]}$ across experiments, with the only variable being the source of the $\nabla G$ corpus. Randomly generated text yielded the poorest performance, while the "locality_prompt" from COUNTERFACT achieved the second-best results, only surpassed by Wikipedia, which produced the best outcomes. These results also serve as reverse validation that the implicit knowledge within the model is embedded in its pre-training data.

Table 2: **Different corpus results for COUNTERFACT.** $T$: Num Edits.

| MEMIT | COUNTERFACT | | | | | | | | | | | | | | | |
| | $T = 200$ | | | | $T = 500$ | | | | $T = 1000$ | | | | $T = 1500$ | | | |
| | Rel. | Gen. | Loc. | Avg. | Rel. | Gen. | Loc. | Avg. | Rel. | Gen. | Loc. | Avg. | Rel. | Gen. | Loc. | Avg. |
| Mistral-7B | | | | | | | | | | | | | | | | |
| Corpus❶ | 0.89 | 0.62 | 0.74 | 0.75 | 0.79 | 0.55 | **0.61** | 0.65 | 0.64 | 0.37 | 0.52 | 0.54 | 0.57 | 0.39 | 0.51 | 0.49 |
| Corpus❷ | **0.90** | **0.62** | 0.73 | 0.75 | 0.76 | 0.53 | 0.60 | 0.63 | 0.62 | 0.33 | 0.50 | 0.48 | 0.54 | 0.36 | 0.49 | 0.46 |
| Corpus❸ | 0.86 | 0.60 | 0.73 | 0.73 | 0.74 | 0.51 | 0.56 | 0.60 | 0.59 | 0.32 | 0.44 | 0.45 | 0.57 | 0.33 | 0.46 | 0.45 |
| Corpus❹ | 0.89 | 0.61 | **0.78** | **0.76** | **0.81** | **0.55** | 0.60 | **0.65** | **0.68** | **0.39** | **0.55** | **0.54** | **0.61** | **0.42** | **0.53** | **0.52** |
| Llama3-8B | | | | | | | | | | | | | | | | |
| Corpus❶ | 0.88 | 0.47 | 0.65 | 0.67 | **0.85** | 0.48 | 0.36 | 0.56 | 0.79 | 0.47 | 0.29 | 0.52 | 0.77 | 0.46 | 0.26 | 0.49 |
| Corpus❷ | **0.91** | 0.48 | 0.66 | 0.68 | 0.85 | 0.50 | 0.40 | 0.58 | 0.79 | 0.47 | 0.30 | 0.52 | 0.74 | 0.44 | 0.27 | 0.48 |
| Corpus❸ | 0.85 | 0.41 | 0.63 | 0.63 | 0.83 | 0.45 | 0.31 | 0.53 | 0.74 | 0.41 | 0.24 | 0.46 | 0.70 | 0.35 | 0.19 | 0.41 |
| Corpus❹ | 0.88 | **0.53** | **0.76** | **0.72** | 0.84 | **0.51** | **0.45** | **0.60** | **0.81** | **0.50** | **0.31** | **0.54** | **0.79** | **0.44** | **0.28** | **0.50** |

## B.7 BASELINE EDITING METHODS

We selected five popular model editing methods as baselines:

- **Fine-Tuning (FT)**, we employ the reimplementation guidelines from Yao et al. (2023). This involves utilizing the Adam optimizer and implementing early stopping to minimize $-\log P_{LM}[*|p]$, while only adjusting $W_{\text{proj}}$.
- **Elastic Weight Consolidation (FT-EWC)** has been shown to effectively mitigate catastrophic forgetting by updating weights based on a Fisher information matrix, which is derived from past edits and scaled by a factor $\lambda$. In line with Wang et al. (2024b), we have chosen to omit the constraints of the $L^{\infty}$ norm in this implementation.
- **MEND** (Mitchell et al., 2022a) adeptly manipulates the gradient of fine-tuned language models by capitalizing on a low-rank decomposition of the gradients, thereby enhancing the accuracy of the editing process. We use the default settings from Yao et al. (2023).
- **ROME** (Meng et al., 2023a) has been previously discussed. In this experiment, we edit the 8th layer, which is regarded as a crucial location for knowledge storage. We utilize second moment statistics $C \propto E[kk^T]$ computed from more than 100,000 samples of hidden states $k$ derived from tokens sampled across all Wikipedia text in context.
- **MEMIT** (Meng et al., 2023b)—the detailed computation process can be found in Appendix B.4. We set $\lambda = 15,000$ to balance the knowledge in the model with the knowledge required for editing. Other settings are consistent with those in ROME.
- **R-Edit** (Gupta et al., 2024a) attributes the suboptimal performance of ROME and MEMIT to the inadequacy of the calculated $k_*$ in representing the subject of the queried knowledge. R-Edit enhances the calculation of $k_*$ in Eqs. 15 and 18 to address this issue.
- **WilKE** (Hu et al., 2024b) argues that different types of knowledge should be distributed across various layers. For each piece of knowledge edited, WilKE first determines the optimal layer for editing and then applies either ROME or MEMIT to perform the edit. Due to the time and computational cost of finding the optimal layer, we restrict the editable layers in this paper to $l = \{5, 6, 7, 8, 9, 10\}$.

- **PRUNE** (Ma et al., 2024) suggests that the key factor influencing sequential editing performance is the condition number of the matrix. PRUNE scales the singular values in $\Delta W_{\text{total}}$ that exceed the maximum singular value of the original model, ensuring that no singular value surpasses a specified threshold. We adhere to the experimental setup outlined in Ma et al. (2024) and scale the larger singular values using the following method:

$$F(\hat{\sigma}_i) = \log_{1.2}(\hat{\sigma}_i) - \log_{1.2}(\max\{\sigma_i\}) + \max\{\sigma_i\}.$$

Table 3: **Different orthogonal method results for COUNTERFACT.** $T$: Num Edits.

| Method | COUNTERFACT | | | | | | | | | | | | | | | |
|---|---|---|---|---|---|---|---|---|---|---|---|---|---|---|---|---|
| | $T=200$ | | | | $T=500$ | | | | $T=1000$ | | | | $T=1500$ | | | |
| | Rel. | Gen. | Loc. | Avg. | Rel. | Gen. | Loc. | Avg. | Rel. | Gen. | Loc. | Avg. | Rel. | Gen. | Loc. | Avg. |
| Mistral-7B | | | | | | | | | | | | | | | | |
| MEMIT | **0.93** | **0.67** | 0.41 | 0.67 | 0.50 | 0.35 | 0.10 | 0.32 | 0.28 | 0.10 | 0.06 | 0.15 | 0.19 | 0.06 | 0.05 | 0.10 |
| ❶Only$\Delta W_{\text{[total]}}$ | 0.91 | 0.54 | 0.77 | 0.74 | 0.79 | 0.53 | 0.55 | 0.62 | 0.61 | 0.37 | 0.51 | 0.50 | 0.55 | 0.37 | 0.46 | 0.46 |
| ❷Only$\nabla G$ | 0.89 | 0.55 | 0.74 | 0.72 | 0.76 | 0.50 | 0.51 | 0.59 | 0.57 | 0.34 | 0.49 | 0.47 | 0.44 | 0.27 | 0.24 | 0.32 |
| ❸Without Eqn.7(26) | 0.89 | 0.59 | 0.77 | 0.75 | 0.78 | 0.56 | 0.56 | 0.63 | 0.58 | 0.37 | 0.52 | 0.49 | 0.49 | 0.36 | 0.49 | 0.44 |
| O-Edit+ | 0.89 | 0.61 | **0.78** | **0.76** | **0.81** | **0.55** | **0.60** | **0.65** | **0.68** | **0.39** | **0.55** | **0.54** | **0.61** | **0.42** | **0.53** | **0.52** |
| Llama3-8B | | | | | | | | | | | | | | | | |
| MEMIT | 0.85 | 0.51 | 0.22 | 0.52 | 0.50 | 0.35 | 0.10 | 0.32 | 0.28 | 0.10 | 0.05 | 0.14 | 0.18 | 0.06 | 0.05 | 0.10 |
| ❶Only$\Delta W_{\text{[total]}}$ | **0.91** | 0.49 | 0.65 | 0.68 | 0.87 | **0.54** | 0.36 | 0.59 | 0.78 | 0.45 | 0.28 | 0.50 | 0.74 | 0.41 | 0.25 | 0.47 |
| ❷Only$\nabla G$ | 0.87 | 0.49 | 0.62 | 0.66 | 0.77 | 0.41 | 0.32 | 0.50 | 0.64 | 0.32 | 0.28 | 0.41 | 0.55 | 0.28 | 0.22 | 0.35 |
| ❸Without Eqn.7(26) | 0.88 | 0.48 | 0.65 | 0.67 | **0.87** | 0.50 | 0.41 | 0.60 | 0.78 | 0.46 | 0.32 | 0.52 | 0.67 | 0.39 | 0.28 | 0.43 |
| O-Edit+ | 0.88 | **0.53** | **0.76** | **0.72** | 0.84 | 0.51 | **0.45** | **0.60** | **0.81** | **0.50** | **0.31** | **0.54** | **0.79** | **0.44** | **0.28** | **0.50** |

### B.8 EXPERIMENTS COMPUTE RESOURCES TIME AND HYPERPARAMETERS

We conducted our experiments using NVIDIA A100 40GB GPUs. For Mistral-7B and LLaMA3-8B, ROME and MEMIT require approximately 35GB of memory and take about 2.5 hours to process 1500 edits. In comparison, O-Edit and O-Edit+ take about 4.5 hours for the same number of edits. The additional computation time is primarily due to the singular value decomposition (SVD) of matrices. For $\nabla G$, its SVD is computed once prior to the first edit, with the $V$ matrix saved for reuse. However, for $\Delta W_{\text{[total]}}$, which is dynamically updated, the SVD must be recomputed after each knowledge edit. On average, computing the SVD for a matrix $W \in \mathbb{R}^{4096 \times 14336}$ takes approximately **4 seconds**, while a single edit using ROME or MEMIT takes around **6 seconds**. For sequential editing, O-Edit and O-Edit+ require only one SVD computation on average per edit, with results significantly surpassing those of traditional methods by several times. See Table 4 for the specific computation times.

Table 4: Computation Time (seconds).

| Method | Datasets | | | |
|---|---|---|---|---|
| | COUNTERFACT-1500 | ZsRE-1500 | RECENT-1200 | WIKICF-400 |
| ROME | 8716 | 8694 | 6917 | 2251 |
| +O-Edit | 13289 | 13961 | 10663 | 4591 |
| +O-Edit+ | 12286 | 12664 | 9451 | 4256 |
| MEMIT | 9122 | 9345 | 7533 | 2614 |
| +O-Edit | 15438 | 15957 | 12640 | 4997 |
| +O-Edit+ | 14766 | 14664 | 10854 | 4651 |

For all experimental settings of O-Edit, we set $\lambda_1$ and $\lambda_2 = 50$. For O-Edit+, we set $\lambda_3 = 2$ for Mistral-7B and $\lambda_3 = 1$ for LLaMA-8B in MEMIT; $\lambda_3 = 2.5$ for both Mistral-7B and LLaMA3-8B in ROME. In the next Section B.9, we conducted detailed ablation experiments and parameter selection experiments to further analyze the impact of hyperparameters on editing performance.

Another potential issue arises when $q$ exceeds the dimensions of the model $(\min(d, d_m))$[5]. In this paper, we have considered 1500 edits. When the number of required edits exceeds this amount, $q$ can be constrained by setting it below a certain threshold to ensure the feasibility of performing additional edits. A smaller threshold for $q$ typically results in more effective edits, while a larger threshold tends to preserve the model's ability to retain unrelated knowledge. However, in general, increasing the number of edits tends to cause greater degradation in the model's performance.

---

[5]The dimension of $W_{proj}$ in both Mistral-7B and LLaMA3-8B is $\mathbb{R}^{4096 \times 14336}$.

Table 5: **Hpyerparameter selection results for O-Edit+.** $T$: Num Edits.

| Method | COUNTERFACT | | | | | | | | | | | | | | | |
|---|---|---|---|---|---|---|---|---|---|---|---|---|---|---|---|---|
| | $T = 200$ | | | | $T = 500$ | | | | $T = 1000$ | | | | $T = 1500$ | | | |
| | Rel. | Gen. | Loc. | Avg. | Rel. | Gen. | Loc. | Avg. | Rel. | Gen. | Loc. | Avg. | Rel. | Gen. | Loc. | Avg. |
| Mistral-7B | | | | | | | | | | | | | | | | |
| **ROME** | 0.72 | 0.53 | 0.31 | 0.52 | 0.30 | 0.18 | 0.14 | 0.21 | 0.28 | 0.10 | 0.06 | 0.15 | 0.27 | 0.13 | 0.05 | 0.13 |
| $\lambda_3 = 1$ | 0.92 | **0.50** | 0.73 | 0.71 | 0.60 | 0.34 | 0.34 | 0.42 | 0.38 | 0.13 | 0.17 | 0.22 | 0.35 | 0.17 | 0.10 | 0.20 |
| $\lambda_3 = 2$ | **0.95** | 0.47 | 0.73 | 0.71 | 0.64 | 0.34 | 0.40 | 0.46 | 0.43 | 0.16 | 0.21 | 0.26 | 0.37 | 0.19 | 0.17 | 0.24 |
| $\lambda_3 = 2.5$ | 0.94 | 0.47 | **0.76** | **0.72** | **0.65** | **0.38** | **0.41** | **0.48** | **0.49** | **0.21** | **0.29** | **0.33** | **0.41** | **0.21** | **0.24** | **0.29** |
| **MEMIT** | **0.93** | 0.67 | 0.41 | 0.67 | 0.50 | 0.35 | 0.10 | 0.32 | 0.28 | 0.10 | 0.06 | 0.15 | 0.19 | 0.06 | 0.05 | 0.10 |
| $\lambda_3 = 1$ | 0.88 | 0.53 | 0.76 | 0.72 | 0.81 | 0.47 | 0.56 | 0.61 | 0.70 | 0.38 | 0.48 | 0.52 | 0.60 | 0.30 | 0.44 | 0.44 |
| $\lambda_3 = 2.5$ | 0.84 | 0.50 | **0.84** | 0.73 | 0.77 | 0.40 | **0.62** | 0.59 | 0.62 | 0.31 | **0.61** | 0.51 | 0.55 | 0.23 | **0.56** | 0.44 |
| $\lambda_3 = 2$ | 0.89 | 0.61 | 0.78 | **0.76** | 0.81 | 0.55 | 0.60 | **0.65** | 0.68 | 0.39 | 0.55 | **0.54** | 0.61 | 0.42 | 0.53 | **0.52** |
| Llama3-8B | | | | | | | | | | | | | | | | |
| **ROME** | 0.75 | 0.48 | 0.14 | 0.46 | 0.69 | 0.45 | 0.05 | 0.40 | 0.75 | 0.46 | 0.02 | 0.41 | 0.47 | 0.28 | 0.02 | 0.31 |
| $\lambda_3 = 1$ | **0.88** | 0.47 | 0.30 | 0.55 | **0.84** | 0.47 | 0.10 | 0.47 | 0.78 | 0.48 | 0.07 | 0.44 | 0.76 | 0.34 | 0.07 | 0.39 |
| $\lambda_3 = 2$ | 0.85 | 0.50 | **0.38** | 0.58 | 0.80 | 0.51 | 0.13 | 0.48 | **0.87** | 0.46 | 0.09 | 0.47 | 0.84 | 0.39 | 0.09 | 0.44 |
| $\lambda_3 = 2.5$ | 0.86 | **0.61** | 0.37 | **0.61** | 0.81 | **0.52** | **0.24** | **0.52** | 0.86 | **0.49** | **0.19** | **0.51** | **0.87** | **0.50** | **0.13** | **0.50** |
| **MEMIT** | 0.85 | 0.51 | 0.22 | 0.52 | 0.50 | 0.35 | 0.10 | 0.32 | 0.28 | 0.10 | 0.05 | 0.14 | 0.18 | 0.06 | 0.05 | 0.10 |
| $\lambda_3 = 0.5$ | 0.88 | 0.47 | 0.61 | 0.65 | 0.84 | 0.47 | 0.38 | 0.56 | 0.78 | 0.48 | 0.30 | 0.52 | 0.76 | **0.46** | 0.25 | 0.49 |
| $\lambda_3 = 2$ | 0.86 | 0.49 | 0.66 | 0.67 | 0.84 | **0.52** | 0.42 | 0.59 | 0.78 | 0.46 | **0.33** | 0.52 | 0.73 | 0.43 | **0.30** | 0.49 |
| $\lambda_3 = 1$ | **0.88** | **0.53** | **0.76** | **0.72** | **0.84** | 0.51 | **0.45** | **0.60** | **0.81** | **0.50** | 0.31 | **0.54** | **0.79** | 0.44 | 0.28 | **0.50** |

## B.9 ABLATION EXPERIMENTS

First, we wanted to see if both $\Delta W_{[\text{total}]}$ and $\nabla G$ contributed effectively. We set up three baselines: ❶ using only $\Delta W_{[\text{total}]}$; ❷ using only $\nabla G$; and ❸ using both $\Delta W_{[\text{total}]}$ and $\nabla G$ without orthogonal processing for $\nabla G$ according to Eq.7(26). The results are shown in Table 3. We observed that while using either $\Delta W_{[\text{total}]}$ or $\nabla G$ alone yielded better results than the original method, their performance was still inferior to using both together. The lack of orthogonalization for $\nabla G$ led to knowledge conflicts within the model, resulting in inferior performance compared to O-Edit+.

**How does the degree of orthogonality between knowledge affect the effectiveness of sequential editing?**

We compared the effects of different hyperparameter selections on editing performance between O-Edit and O-Edit+, as shown in Tables 5 and 6. In O-Edit+, two noteworthy phenomena were observed. First, MEMIT's $\lambda_3$ is smaller than that of ROME due to ROME's stronger constraints, which can degrade the performance of unrelated knowledge (**Loc.**) during sequential editing. Consequently, we opted for a larger $\lambda_3 = 2.5$ to mitigate ROME's influence. Second, while a smaller $\lambda_3$ improves performance with MEMIT, it still negatively impacts unrelated knowledge, and a larger $\lambda_3$ affects the editing effect (**Rel., Gen.**). Therefore, selecting an appropriately sized $\lambda_3$ is crucial for optimal overall editing performance.

In the O-Edit setting, we compared the editing performance under four different settings. The results showed that stronger constraints led to better outcomes, as $\lambda_1$ and $\lambda_2$ effectively controlled the correlation between different edits. Larger $\lambda_1$ values resulted in smaller correlations between edits, while larger $\lambda_2$ values reduced the correlation between edited and implicit knowledge within the model.

## B.10 EDITING DATASETS AND EXTRA METRICS

- **ZsRE** question answering task (Levy et al., 2017) was first used for factual knowledge evaluation by (Cao et al., 2021), later being extended and adopted by (Mitchell et al., 2022a). We conduct the experiment using the version provided by (Yao et al., 2023) in EasyEdit[6]. Figure 8 shows examples from ZsRE.

- **COUNTERFACT** is designed to enable distinction between superficial changes in model word choices from specific and generalized changes in underlying factual knowledge. Figure 9 shows examples from COUNTERFACT.

---

[6]https://github.com/zjunlp/EasyEdit

- **RECENT** Zhang et al. (2024a) is a dataset that specifically focuses on triplets that have been recently inserted into WIKIDATA after July 2022. Consequently, this dataset enables us to create insertion edit requests for models that were trained prior to the introduction of these facts, thereby simulating scenarios where an outdated model meets the new world knowledge. We utilize the original datasets provided by the authors and split them into training and testing sets.

- **WIKICF**: Since tail entities are often not captured by models, and therefore are not suitable for testing modification edits, the authors (Zhang et al., 2024a) collect triplets about popular entities, where the subject corresponds to one of the top-viewed pages in Wikipedia. They also collect a dataset by randomly sampling entities from Wikidata, which we use as the training set, and the WikiDataCounterFact as the test set.

- **Portability (Port.)**: Knowledge is not isolated, and solely changing the given knowledge is not enough for downstream use. When the knowledge is corrected, the model is supposed to reason about the downstream effects of the correction. Here, we follow previous work (Cohen et al., 2023; Zhang et al., 2024a) to evaluate whether the edited model can address the implications of an edit for real-world applications.

- **Subject Aliasing (Alg.):** The editing of one subject should not vary from its expression. Wikidata maintains a set of aliases for every entity. Hence, here, we follow Cohen et al. (2023); Yao et al. (2023) to replace the question's subject with an alias or synonym to evaluate the post-edited model's performance on other descriptions of the subject.

- **Compositionality and Reasoning (Res.):** This requires the post-edit model to conduct reasoning with the changed facts. For example, when we change the current president of the U.S. from Donald Trump to Joe Biden, the answer to the question "Who is the First Lady of the United States?" should also be changed.

- **Forgetfulness (Fog.):** This evaluates whether the post-edit model retains the original objects in one-to-many relationships. we follow Zhang et al. (2024a) to evaluate this metric.

- **Logical Generalization(Lgn.):** These are the changes that are semantically related to the modified fact and expected to change by the edit; they were indeed modified. For example, as mentioned by (Yao et al., 2023), when the fact of $(s, r, o)$ is changed, the reversed relation of the knowledge $(o, \hat{r}, s)$ should also be changed.

Table 6: **Hpyerparameter selection results for O-Edit.** $T$: Num Edits.

| Method | COUNTERFACT | | | | | | | | | | | | | | | |
| | $T = 200$ | | | | $T = 500$ | | | | $T = 1000$ | | | | $T = 1500$ | | | |
| | Rel. | Gen. | Loc. | Avg. | Rel. | Gen. | Loc. | Avg. | Rel. | Gen. | Loc. | Avg. | Rel. | Gen. | Loc. | Avg. |
| Mistral-7B | | | | | | | | | | | | | | | | |
| **MEMIT** | 0.93 | **0.67** | 0.41 | 0.67 | 0.50 | 0.35 | 0.10 | 0.32 | 0.28 | 0.10 | 0.06 | 0.15 | 0.19 | 0.06 | 0.05 | 0.10 |
| $\lambda_1, \lambda_2 = 1$ | **0.95** | 0.66 | 0.52 | 0.68 | 0.74 | 0.36 | 0.29 | 0.46 | 0.40 | 0.24 | 0.11 | 0.25 | 0.39 | 0.19 | 0.08 | 0.22 |
| $\lambda_1, \lambda_2 = 10$ | 0.93 | 0.62 | 0.54 | 0.70 | **0.89** | 0.50 | 0.35 | 0.58 | 0.54 | 0.26 | 0.18 | 0.31 | 0.45 | 0.22 | 0.10 | 0.26 |
| $\lambda_1, \lambda_2 = 20$ | 0.93 | 0.53 | 0.62 | 0.69 | 0.87 | 0.52 | 0.36 | 0.58 | 0.64 | 0.39 | 0.24 | 0.42 | 0.47 | 0.26 | 0.13 | 0.29 |
| $\lambda_1, \lambda_2 = 50$ | 0.93 | 0.55 | **0.65** | **0.71** | 0.86 | **0.53** | **0.45** | **0.61** | **0.72** | **0.47** | **0.34** | **0.51** | **0.51** | **0.33** | **0.18** | **0.34** |
| Llama3-8B | | | | | | | | | | | | | | | | |
| **MEMIT** | 0.85 | 0.51 | 0.22 | 0.52 | 0.50 | 0.35 | 0.10 | 0.32 | 0.28 | 0.10 | 0.05 | 0.14 | 0.18 | 0.06 | 0.05 | 0.10 |
| $\lambda_1, \lambda_2 = 1$ | 0.96 | 0.52 | 0.43 | 0.63 | 0.83 | 0.59 | 0.16 | 0.52 | 0.62 | 0.49 | 0.08 | 0.40 | 0.35 | 0.27 | 0.08 | 0.23 |
| $\lambda_1, \lambda_2 = 10$ | **0.97** | 0.47 | 0.53 | 0.65 | 0.90 | 0.55 | 0.35 | 0.60 | 0.72 | 0.51 | 0.18 | 0.47 | 0.45 | 0.33 | 0.10 | 0.29 |
| $\lambda_1, \lambda_2 = 20$ | 0.96 | 0.42 | 0.57 | 0.65 | **0.90** | **0.54** | 0.41 | 0.61 | **0.75** | **0.52** | 0.22 | 0.49 | 0.45 | 0.35 | 0.15 | 0.31 |
| $\lambda_1, \lambda_2 = 50$ | 0.93 | **0.55** | **0.64** | **0.71** | 0.86 | 0.53 | **0.44** | **0.61** | 0.72 | 0.47 | **0.33** | **0.51** | **0.55** | **0.40** | **0.27** | **0.41** |

## B.11 DOWNSTREAM TASKS SETTINGS

To explore the side effects of sequential model editing on the general abilities of LLMs, four representative tasks with corresponding datasets were adopted for assessment, including: **Commonsense Reasoning** on the **SIQA** (Sap et al., 2019), which is a benchmark for testing social commonsense intelligence. **Content Analysis** on the **LAMBADA** (Paperno et al., 2016), which is a collection of narrative paragraphs that requires computational models to track information across a broader discourse. **Question Answering** on the **CommonsenseQA** (Talmor et al., 2019), it requires the model be capable of making reasonable inferences under given common sense conditions. **MATH** on the **GSM8K** (Cobbe et al., 2021), a dataset of 8.5K high-quality linguistically diverse grade

```
{
    "subject": "Watts Humphrey",
    "src": "What university did Watts Humphrey attend?",
    "pred": "Trinity College",
    "rephrase": "What university did Watts Humphrey take part in?",
    "alt": "University of Michigan",
    "answers": [
        "Illinois Institute of Technology"
    ],
    "loc": "nq question: who played desmond doss father in hacksaw ridge",
    "loc_ans": "Hugo Weaving",
    "cond": "Trinity College >> University of Michigan || What university did Watts Humphrey attend?"
}
```

Figure 8: Sample of ZsRE Dataset

```
{
    "case_id": 1,
    "prompt": "The official religion of Edwin of Northumbria is",
    "target_new": "Islam",
    "subject": "Edwin of Northumbria",
    "ground_truth": "Christianity",
    "rephrase_prompt": "The school chiefly served tribal girls of Dang.
                        Edwin of Northumbria follows the religion
                        of",
    "locality_prompt": "Fine Young Cannibals was founded in",
    "locality_ground_truth": "Birmingham"
}
```

Figure 9: Sample of COUNTERFACT Dataset

school math word problems. The prompts for each downstream task were illustrated in Table 7. We utilized OpenCompass[7] to conduct our evaluations.

---

[7] https://github.com/open-compass/opencompass

Table 7: The prompts to LLMs for evaluating their zero-shot performance on these general tasks.

| Task | Prompt |
|---|---|
| SIQA | prompt= "{question} A. {A} B. {B} C. {C} Answer:" |
| LAMBADA | prompt= "Please complete the following sentence: {sentence}" |
| CommonsenseQA | prompt= "{question} A. {A} B. {B} C. {C} D. {D} E. {E} Answer:" |
| GSM8K | prompt = " Question: {question} Let's think step by step. Answer:" |

Table 8: **The results of different method with similar $\|\Delta W_{[\text{total}]}\|_2$.** $T$: Num Edits.

| Method | COUNTERFACT | | | | | | | | | | | | | | | |
|---|---|---|---|---|---|---|---|---|---|---|---|---|---|---|---|---|
| | $T = 200$ | | | | $T = 500$ | | | | $T = 1000$ | | | | $T = 1500$ | | | |
| | Rel. | Gen. | Loc. | Avg. | Rel. | Gen. | Loc. | Avg. | Rel. | Gen. | Loc. | Avg. | Rel. | Gen. | Loc. | Avg. |
| Mistral-7B | | | | | | | | | | | | | | | | |
| MEMIT | **0.93** | 0.67 | 0.41 | 0.67 | 0.50 | 0.35 | 0.10 | 0.32 | 0.28 | 0.10 | 0.06 | 0.15 | 0.19 | 0.06 | 0.05 | 0.10 |
| Method ❶ | 0.88 | 0.50 | 0.70 | 0.69 | 0.41 | 0.22 | 0.44 | 0.36 | 0.27 | 0.14 | 0.11 | 0.17 | 0.20 | 0.08 | 0.09 | 0.12 |
| Method ❷ | 0.83 | 0.44 | 0.67 | 0.64 | 0.57 | 0.34 | 0.31 | 0.40 | 0.35 | 0.21 | 0.08 | 0.21 | 0.22 | 0.13 | 0.04 | 0.13 |
| Method ❸ | 0.86 | 0.47 | 0.61 | 0.64 | 0.60 | 0.37 | 0.30 | 0.42 | 0.31 | 0.17 | 0.11 | 0.20 | 0.18 | 0.10 | 0.06 | 0.11 |
| Method ❹ | 0.84 | 0.55 | 0.61 | 0.67 | 0.57 | 0.33 | 0.31 | 0.40 | 0.29 | 0.19 | 0.11 | 0.20 | 0.21 | 0.11 | 0.05 | 0.12 |
| +O-Edit+ | 0.89 | **0.61** | **0.78** | **0.76** | **0.81** | **0.55** | **0.60** | **0.65** | **0.68** | **0.39** | **0.55** | **0.54** | **0.61** | **0.42** | **0.53** | **0.52** |
| Llama3-8B | | | | | | | | | | | | | | | | |
| MEMIT | 0.85 | 0.51 | 0.22 | 0.52 | 0.50 | 0.35 | 0.10 | 0.32 | 0.28 | 0.10 | 0.05 | 0.14 | 0.18 | 0.06 | 0.05 | 0.10 |
| Method ❶ | 0.74 | 0.33 | 0.58 | 0.32 | 0.32 | 0.11 | 0.51 | 0.31 | 0.24 | 0.08 | 0.34 | 0.22 | 0.13 | 0.07 | 0.18 | 0.12 |
| Method ❷ | 0.83 | 0.50 | 0.24 | 0.52 | 0.72 | 0.37 | 0.08 | 0.39 | 0.44 | 0.19 | 0.08 | 0.23 | 0.40 | 0.13 | 0.08 | 0.20 |
| Method ❸ | 0.82 | 0.49 | 0.28 | 0.53 | 0.72 | 0.35 | 0.08 | 0.38 | 0.46 | 0.21 | 0.03 | 0.23 | 0.32 | 0.17 | 0.02 | 0.17 |
| Method ❹ | 0.77 | 0.41 | 0.44 | 0.54 | 0.69 | 0.32 | 0.06 | 0.36 | 0.47 | 0.23 | 0.31 | 0.33 | 0.37 | 0.15 | 0.09 | 0.21 |
| +O-Edit+ | **0.88** | **0.53** | **0.76** | **0.72** | **0.84** | **0.51** | **0.45** | **0.60** | **0.81** | **0.50** | **0.31** | **0.54** | **0.79** | **0.44** | **0.28** | **0.50** |

## B.12 FURTHER EXPERIMENT AND DISCUSSION

**Can any method of reducing $\Delta W_{[\text{total}]}$ improve the ability of sequential editing?**

Hu et al. (2024b) posits that $\|\Delta W_{[\text{total}]}\|_2$ is a key determinant of sequential editing, referred to as **"toxicity"**. A higher $\|\Delta W_{[\text{total}]}\|_2$ imposes greater constraints on sequential editing performance. O-Edit+ effectively reduces $\|\Delta W_{[\text{total}]}\|_2$ by diminishing projections in specific subspaces. Consequently, a plausible hypothesis is that any method capable of reducing $\|\Delta W_{[\text{total}]}\|_2$ could potentially enhance sequential editing performance. To evaluate this hypothesis, we compare O-Edit+ with four methods on COUNTERFACT: ❶ reducing the number of training steps to decrease $\|v_* - Wk_*\|_2$, thereby reducing $\|\Delta W_{[\text{total}]}\|_2$ with each edit; ❷ randomly deleting some values in the update parameters, setting them to zero; ❸ randomly selecting a set of orthogonal subspaces and removing the projection of $\Delta W_i$ onto them; ❹ multiplying the $\Delta W$ obtained by the original method by a coefficient $\eta$ that is less than 1, updating the matrix as $\Delta W = \eta \cdot \Delta W$. We adjust the hyperparameters to ensure that the $\|\Delta W_{[\text{total}]}\|_2$ generated by these methods approximates that of O-Edit+. As shown in Table 8, although these five methods yield a similar $\|\Delta W_{[\text{total}]}\|_2$, the first four fail to achieve effective sequential editing. This indicates that while reducing $\|\Delta W_{[\text{total}]}\|_2$ is a necessary but not sufficient condition for successful sequential editing, choosing the correct projection space to ensure minimal impact between knowledge is the key to the success of ours.

**Theoretical analysis**

Considering MEMIT, we derive from the equations $\Delta W_{\text{[total]}} = \sum_{i=1}^{n} \Delta W_i$ and $\Delta W_{\text{unrelated}} = \Delta W_{\text{[total]}} - \Delta W_j$ that:

$$\|\Delta W_{\text{unrelated}} \cdot k_j\|_2 = \|(\Delta W_{\text{[total]}} - \Delta W_j) \cdot k_j\|_2$$

$$= \left\| \sum_{i=1, i \neq j}^{n} \Delta W_i \cdot k_j \right\|_2$$

$$= \left\| \sum_{i=1; i \neq j}^{n} R_i k_{*;i}^T (\lambda K K^T + k_{*;i} k_{*;i}^T)^{-1} \cdot k_j \right\|_2$$

$$= \sum_{i=1; i \neq j}^{n} \left( R_i k_{*;i}^T (\lambda K K^T + k_{*;i} k_{*;i}^T)^{-1} k_j \right)^T \sum_{i=1; i \neq j}^{n} R_i k_{*;i}^T (\lambda K K^T + k_{*;i} k_{*;i}^T)^{-1} k_j$$

$$= \sum_{i=1; i \neq j}^{n} k_j^T \left( (\lambda K K^T + k_{*;i} k_{*;i}^T)^{-1} \right)^T k_{*;i} R_i^T \sum_{i=1; i \neq j}^{n} R_i k_{*;i}^T (K K^T + k_{*;i} k_{*;i}^T)^{-1} k_j$$

$$(29)$$

Since $R$ is a column vector, $R^T$ is a row vector. For any $R_n$ and $R_m$ where $n \neq m$, the updates in O-Edit and O-Edit+ aim to ensure that each update matrix $\Delta W$ is orthogonal in the column space, leading to $R_n^T \cdot R_m \rightarrow 0$. Consequently, the value of Eqn. 29 is smaller than that of MEMIT.

**Differences and Similarities with Hu et al. (2024a)**

From the perspective of activating $\|\Delta W_{\text{unrelated}} \cdot k_j\|_2$, (Hu et al., 2024a) emphasizes the reduction of this metric's activation value through orthogonal row space. They aim to achieve smaller activation values using the expression $\sum_{i=1; i \neq j}^{n} k_{*;i}^T (\lambda K K^T + k_{*;i} k_{*;i}^T)^{-1} k_j \rightarrow 0$. However, since the variables $K$ and $k_*$ are predetermined, their orthogonality cannot be optimized through training methods. To address this, they suggest selecting bottom layers with lower row orthogonality. Yet, this method undermines the extensibility of editing techniques, as knowledge is not solely stored in the lower layers of the model (Li et al., 2024; Meng et al., 2023a; Geva et al., 2021; 2022).

In contrast, O-Edit and O-Edit+ tackle this issue by focusing on orthogonal column space, providing a practical algorithm that supports multiple consecutive edits. These methods can achieve column space orthogonality between update matrices at any layer, effectively reducing $\|\Delta W_{\text{unrelated}} \cdot k_j\|_2$ and facilitating expansion to multi-layer editing.

## B.13 FURTHER EDITING DATASETS RESULTS

Table 9: **Main editing results for ZsRE.** $T$: Num Edits.

| Method | ZsRE | | | | | | | | | | | | | | | |
| --- | --- | --- | --- | --- | --- | --- | --- | --- | --- | --- | --- | --- | --- | --- | --- | --- |
| | $T = 200$ | | | | $T = 500$ | | | | $T = 1000$ | | | | $T = 1500$ | | | |
| | Rel. | Gen. | Loc. | Avg. | Rel. | Gen. | Loc. | Avg. | Rel. | Gen. | Loc. | Avg. | Rel. | Gen. | Loc. | Avg. |
| **Mistral-7B** | | | | | | | | | | | | | | | | |
| **ROME** | 0.82 | 0.41 | 0.38 | 0.53 | 0.32 | 0.22 | 0.08 | 0.20 | 0.30 | 0.17 | 0.06 | 0.18 | 0.31 | 0.15 | 0.06 | 0.17 |
| **+R-Edit** | 0.95 | 0.49 | 0.47 | 0.64 | 0.27 | 0.18 | 0.08 | 0.17 | 0.31 | 0.13 | 0.05 | 0.16 | 0.31 | 0.15 | 0.06 | 0.17 |
| **+WilKE** | 0.88 | 0.44 | 0.43 | 0.58 | 0.41 | 0.27 | 0.10 | 0.26 | 0.27 | 0.17 | 0.06 | 0.17 | 0.29 | 0.19 | 0.05 | 0.17 |
| **+PRUNE** | 0.92 | 0.30 | 0.82 | 0.68 | 0.77 | 0.32 | 0.53 | 0.54 | 0.36 | 0.19 | 0.34 | 0.30 | 0.33 | 0.21 | **0.27** | 0.27 |
| **+O-Edit** | 0.99 | 0.42 | 0.73 | 0.71 | 0.77 | 0.41 | 0.51 | 0.49 | 0.45 | 0.18 | 0.29 | 0.31 | 0.35 | 0.20 | 0.20 | 0.25 |
| **+O-Edit+** | **0.99** | **0.46** | **0.75** | **0.73** | **0.80** | **0.45** | **0.51** | **0.60** | **0.68** | **0.42** | **0.32** | **0.47** | **0.43** | **0.16** | 0.25 | **0.28** |
| **MEMIT** | 0.95 | 0.50 | 0.38 | 0.61 | 0.52 | 0.37 | 0.14 | 0.34 | 0.31 | 0.20 | 0.06 | 0.19 | 0.24 | 0.10 | 0.06 | 0.13 |
| **+R-Edit** | 0.96 | 0.49 | 0.41 | 0.62 | 0.40 | 0.19 | 0.40 | 0.44 | 0.32 | 0.22 | 0.06 | 0.20 | 0.26 | 0.16 | 0.07 | 0.16 |
| **+WilKE** | **0.99** | 0.50 | 0.47 | 0.65 | 0.75 | 0.47 | 0.23 | 0.48 | 0.25 | 0.20 | 0.06 | 0.17 | 0.28 | 0.15 | 0.04 | 0.16 |
| **+PRUNE** | 0.83 | **0.53** | 0.47 | 0.61 | 0.76 | **0.52** | 0.29 | 0.52 | 0.65 | **0.45** | 0.22 | 0.44 | 0.43 | 0.27 | 0.12 | 0.27 |
| **+O-Edit** | 0.97 | 0.40 | 0.65 | 0.67 | **0.88** | 0.42 | 0.43 | 0.57 | **0.76** | 0.41 | 0.39 | **0.52** | 0.61 | **0.33** | 0.18 | 0.37 |
| **+O-Edit+** | 0.94 | 0.33 | **0.80** | **0.69** | 0.82 | 0.33 | **0.60** | **0.58** | 0.69 | 0.31 | **0.54** | 0.51 | 0.60 | 0.26 | **0.51** | **0.43** |
| **Llama3-8B** | | | | | | | | | | | | | | | | |
| **ROME** | 0.84 | 0.63 | 0.23 | 0.56 | 0.69 | **0.62** | 0.03 | 0.44 | 0.73 | **0.60** | 0.03 | 0.45 | 0.74 | **0.63** | 0.02 | 0.46 |
| **+R-Edit** | 0.86 | 0.51 | 0.38 | 0.58 | 0.62 | 0.57 | 0.10 | 0.43 | 0.56 | 0.47 | 0.01 | 0.35 | 0.56 | 0.47 | 0.02 | 0.35 |
| **+WilKE** | 0.75 | 0.37 | 0.28 | 0.47 | 0.50 | 0.38 | 0.05 | 0.31 | 0.60 | 0.50 | 0.02 | 0.37 | 0.66 | 0.55 | 0.01 | 0.40 |
| **+PRUNE** | 0.90 | 0.57 | 0.33 | 0.60 | 0.77 | 0.50 | 0.24 | 0.50 | 0.83 | 0.41 | 0.21 | 0.48 | 0.79 | 0.36 | 0.18 | 0.44 |
| **+O-Edit** | **0.94** | **0.66** | 0.51 | **0.70** | 0.77 | 0.51 | 0.22 | 0.50 | 0.78 | 0.47 | 0.16 | 0.47 | 0.77 | 0.48 | 0.14 | 0.46 |
| **+O-Edit+** | 0.91 | 0.47 | **0.55** | 0.52 | **0.82** | 0.46 | **0.27** | **0.52** | **0.84** | 0.49 | **0.25** | **0.53** | **0.82** | 0.42 | **0.24** | **0.49** |
| **MEMIT** | 0.93 | **0.63** | 0.30 | 0.62 | 0.75 | 0.65 | 0.03 | 0.48 | 0.53 | 0.40 | 0.04 | 0.32 | 0.33 | 0.23 | 0.04 | 0.20 |
| **+R-Edit** | 0.94 | 0.62 | 0.25 | 0.60 | 0.82 | **0.69** | 0.10 | 0.53 | 0.65 | 0.55 | 0.06 | 0.42 | 0.52 | 0.41 | 0.03 | 0.32 |
| **+ WilKE** | **0.98** | 0.42 | **0.70** | **0.70** | 0.78 | 0.65 | 0.10 | 0.51 | 0.61 | 0.50 | 0.07 | 0.40 | 0.52 | **0.42** | 0.05 | 0.33 |
| **+PRUNE** | 0.97 | 0.56 | 0.50 | 0.67 | 0.87 | 0.60 | 0.43 | 0.63 | 0.56 | 0.34 | 0.40 | 0.43 | 0.46 | 0.30 | 0.29 | 0.35 |
| **+O-Edit** | 0.96 | 0.42 | 0.52 | 0.63 | **0.90** | 0.49 | 0.41 | **0.60** | **0.77** | 0.51 | 0.32 | **0.53** | 0.55 | 0.40 | 0.27 | 0.40 |
| **+O-Edit+** | 0.97 | 0.40 | 0.59 | 0.65 | 0.85 | 0.37 | **0.46** | 0.56 | 0.73 | 0.32 | **0.38** | 0.48 | **0.65** | 0.29 | **0.36** | **0.43** |

Table 10: **Main editing results for COUNTERFACT-Portability.** $T$: Num Edits.

| Method | COUNTERFACT-Portability | | | |
|---|---|---|---|---|
| | $T = 200$ | $T = 500$ | $T = 1000$ | $T = 1500$ |
| | **Por.** | **Por.** | **Por.** | **Por.** |
| **Mistral−7B** | | | | |
| **ROME** | 0.48 | 0.04 | 0.01 | 0.01 |
| **+R-Edit** | 0.47 | 0.02 | 0.02 | 0.01 |
| **+WilKE** | 0.41 | 0.2 | 0.2 | 0.02 |
| **+PRUNE** | 0.52 | 0.46 | 0.38 | 0.32 |
| **+O-Edit** | 0.52 | 0.46 | 0.41 | 0.39 |
| **+O-Edit+** | **0.53** | **0.52** | **0.50** | **0.50** |
| **MEMIT** | 0.48 | 0.26 | 0.01 | 0.01 |
| **+R-Edit** | 0.44 | 0.27 | 0.02 | 0.01 |
| **+WilKE** | 0.44 | 0.10 | 0.02 | 0.01 |
| **+PRUNE** | 0.52 | 0.45 | 0.32 | 0.26 |
| **+O-Edit** | 0.51 | 0.45 | 0.38 | 0.34 |
| **+O-Edit+** | **0.53** | **0.53** | **0.53** | **0.51** |
| **Llama3−8B** | | | | |
| **ROME** | 0.26 | 0.07 | 0.01 | 0.01 |
| **+R-Edit** | 0.25 | 0.07 | 0.02 | 0.01 |
| **+WilKE** | 0.24 | 0.07 | 0.02 | 0.02 |
| **+PRUNE** | 0.43 | 0.37 | 0.31 | 0.28 |
| **+O-Edit** | 0.42 | 0.40 | 0.37 | 0.33 |
| **+O-Edit+** | **0.45** | **0.44** | **0.39** | **0.35** |
| **MEMIT** | 0.24 | 0.02 | 0.02 | 0.02 |
| **+R-Edit** | 0.23 | 0.02 | 0.02 | 0.01 |
| **+WilKE** | 0.24 | 0.12 | 0.02 | 0.02 |
| **+PRUNE** | 0.44 | 0.42 | 0.32 | 0.29 |
| **+O-Edit** | 0.45 | 0.42 | 0.39 | 0.39 |
| **+O-Edit+** | **0.49** | **0.48** | **0.46** | **0.45** |

Table 11: **Main editing results for RECENT.** $T$: Num Edits.

| Method | RECENT | | | | | | | | | | | | | | | |
|---|---|---|---|---|---|---|---|---|---|---|---|---|---|---|---|---|
| | $T = 200$ | | | | $T = 500$ | | | | $T = 1000$ | | | | $T = 1200$ | | | |
| | Rel. | Fog. | Alg. | Avg. | Rel. | Fog. | Alg. | Avg. | Rel. | Fog. | Alg. | Avg. | Rel. | Fog. | Alg. | Avg. |
| Mistral-7B | | | | | | | | | | | | | | | | |
| ROME | 0.39 | 0.33 | 0.35 | 0.35 | 0.21 | 0.03 | 0.18 | 0.14 | 0.06 | 0.03 | 0.06 | 0.05 | 0.02 | 0.01 | 0.01 | 0.01 |
| +RRUNE | 0.69 | 0.43 | 0.45 | 0.52 | 0.53 | 0.27 | 0.28 | 0.36 | 0.26 | 0.27 | 0.23 | 0.25 | 0.24 | 0.23 | 0.22 | 0.23 |
| +O-Edit | **0.82** | 0.47 | 0.58 | 0.62 | 0.67 | 0.37 | 0.44 | 0.49 | 0.36 | 0.32 | 0.28 | 0.32 | 0.33 | 0.23 | 0.27 | 0.27 |
| +O-Edit+ | 0.80 | **0.51** | **0.61** | **0.64** | 0.67 | **0.44** | **0.53** | **0.54** | **0.46** | **0.36** | **0.35** | **0.39** | **0.42** | **0.31** | **0.30** | **0.34** |
| MEMIT | 0.82 | 0.48 | 0.67 | 0.66 | 0.16 | 0.02 | 0.15 | 0.11 | 0.08 | 0.00 | 0.07 | 0.05 | 0.07 | 0.00 | 0.05 | 0.04 |
| +RRUNE | 0.86 | 0.60 | 0.68 | 0.71 | 0.74 | 0.45 | 0.55 | 0.58 | 0.57 | 0.37 | 0.52 | 0.48 | 0.46 | 0.31 | 0.40 | 0.39 |
| +O-Edit | 0.88 | 0.58 | 0.64 | 0.70 | **0.79** | 0.50 | 0.61 | 0.63 | 0.62 | 0.40 | 0.51 | 0.51 | 0.57 | 0.35 | 0.44 | 0.45 |
| +O-Edit+ | **0.89** | **0.64** | **0.68** | **0.74** | 0.78 | **0.56** | **0.65** | **0.66** | **0.67** | **0.47** | **0.60** | **0.58** | **0.60** | **0.41** | **0.53** | **0.51** |
| Llama3-8B | | | | | | | | | | | | | | | | |
| ROME | 0.36 | 0.15 | 0.30 | 0.27 | 0.22 | 0.03 | 0.15 | 0.13 | 0.18 | 0.05 | 0.10 | 0.11 | 0.04 | 0.03 | 0.04 | 0.04 |
| +RRUNE | 0.78 | 0.45 | 0.50 | 0.57 | 0.52 | 0.27 | 0.39 | 0.39 | 0.33 | 0.19 | 0.24 | 0.25 | 0.27 | 0.15 | 0.18 | 0.20 |
| +O-Edit | 0.77 | 0.45 | 0.52 | 0.58 | 0.57 | **0.33** | 0.41 | 0.43 | **0.45** | **0.28** | 0.31 | 0.34 | 0.40 | 0.24 | 0.27 | 0.30 |
| +O-Edit+ | **0.78** | 0.45 | **0.56** | **0.60** | **0.57** | 0.31 | **0.48** | **0.45** | 0.44 | 0.27 | **0.33** | **0.34** | **0.41** | **0.25** | **0.30** | **0.32** |
| MEMIT | 0.52 | 0.15 | 0.41 | 0.36 | 0.21 | 0.03 | 0.18 | 0.14 | 0.16 | 0.01 | 0.11 | 0.09 | 0.11 | 0.01 | 0.05 | 0.07 |
| +RRUNE | 0.88 | 0.50 | 0.68 | 0.68 | 0.72 | 0.42 | 0.58 | 0.58 | 0.47 | 0.32 | 0.31 | 0.36 | 0.36 | 0.27 | 0.39 | 0.34 |
| +O-Edit | 0.86 | 0.54 | 0.64 | 0.68 | 0.78 | **0.51** | 0.60 | 0.63 | **0.69** | **0.45** | **0.55** | **0.56** | **0.61** | 0.33 | **0.47** | **0.47** |
| +O-Edit+ | **0.91** | 0.54 | 0.66 | 0.70 | 0.78 | 0.46 | **0.64** | 0.63 | 0.57 | 0.43 | 0.46 | 0.49 | 0.55 | **0.36** | 0.44 | 0.45 |

Table 12: **Main editing results for WIKICF.** $T$: Num Edits.

| Method | WIKICF | | | | | | | | | | | | | | | |
|---|---|---|---|---|---|---|---|---|---|---|---|---|---|---|---|---|
| | $T = 50$ | | | | $T = 100$ | | | | $T = 200$ | | | | $T = 400$ | | | |
| | Rel. | Res. | Lgn. | Avg. | Rel. | Res. | Lgn. | Avg. | Rel. | Res. | Lgn. | Avg. | Rel. | Res. | Lgn. | Avg. |
| Mistral-7B | | | | | | | | | | | | | | | | |
| ROME | 0.81 | 0.56 | 0.54 | 0.63 | 0.35 | 0.42 | 0.30 | 0.35 | 0.18 | 0.06 | 0.18 | 0.12 | 0.12 | 0.03 | 0.06 | 0.07 |
| +RRUNE | 0.82 | 0.54 | 0.60 | 0.65 | 0.67 | 0.55 | 0.54 | 0.58 | 0.48 | 0.51 | 0.50 | 0.49 | 0.42 | 0.44 | 0.39 | 0.41 |
| +O-Edit | 0.82 | 0.62 | **0.68** | **0.71** | 0.67 | 0.58 | 0.61 | 0.62 | 0.54 | 0.53 | 0.59 | 0.55 | 0.48 | 0.47 | 0.44 | 0.46 |
| +O-Edit+ | **0.82** | 0.62 | 0.66 | 0.70 | **0.73** | **0.62** | **0.65** | **0.66** | **0.61** | **0.60** | **0.66** | **0.62** | **0.60** | **0.60** | **0.54** | **0.65** |
| MEMIT | 0.87 | 0.74 | 0.72 | 0.77 | 0.66 | 0.42 | 0.48 | 0.52 | 0.26 | 0.06 | 0.21 | 0.17 | 0.13 | 0.02 | 0.03 | 0.06 |
| +RRUNE | 0.90 | 0.60 | 0.61 | 0.70 | 0.70 | 0.42 | 0.55 | 0.55 | 0.54 | 0.44 | 0.48 | 0.48 | 0.39 | 0.42 | 0.32 | 0.37 |
| +O-Edit | **0.92** | 0.60 | 0.64 | 0.72 | **0.77** | 0.47 | 0.61 | 0.61 | 0.54 | 0.65 | 0.48 | 0.55 | 0.43 | 0.40 | 0.30 | 0.37 |
| +O-Edit+ | 0.84 | **0.66** | **0.69** | **0.73** | 0.73 | **0.66** | 0.60 | **0.62** | **0.62** | **0.72** | **0.65** | **0.66** | **0.58** | **0.48** | **0.48** | **0.51** |
| Llama3-8B | | | | | | | | | | | | | | | | |
| ROME | 0.80 | 0.52 | 0.48 | 0.60 | 0.29 | 0.46 | 0.21 | 0.32 | 0.32 | 0.04 | 0.21 | 0.21 | 0.28 | 0.02 | 0.00 | 0.10 |
| +RRUNE | 0.78 | 0.56 | 0.50 | 0.61 | 0.53 | 0.46 | 0.42 | 0.47 | 0.45 | 0.35 | 0.38 | 0.39 | 0.42 | 0.31 | 0.27 | 0.33 |
| +O-Edit | **0.80** | **0.55** | 0.54 | 0.60 | 0.55 | 0.44 | 0.44 | 0.47 | 0.50 | 0.36 | 0.39 | 0.41 | 0.46 | 0.35 | 0.31 | 0.37 |
| +O-Edit+ | 0.77 | 0.54 | **0.54** | **0.62** | **0.61** | **0.48** | **0.48** | **0.52** | **0.56** | **0.40** | **0.45** | **0.47** | **0.52** | **0.40** | **0.36** | **0.42** |
| MEMIT | 0.75 | 0.52 | 0.48 | 0.58 | 0.57 | 0.24 | 0.24 | 0.35 | 0.31 | 0.04 | 0.06 | 0.13 | 0.28 | 0.02 | 0.00 | 0.10 |
| +RRUNE | 0.80 | 0.68 | 0.62 | 0.70 | 0.74 | 0.48 | 0.62 | 0.61 | 0.60 | 0.52 | 0.47 | 0.53 | 0.55 | 0.36 | 0.39 | 0.43 |
| +O-Edit | 0.80 | 0.70 | 0.60 | 0.70 | 0.72 | 0.50 | 0.64 | 0.62 | 0.65 | 0.51 | 0.50 | 0.55 | 0.61 | 0.40 | 0.41 | 0.47 |
| +O-Edit+ | **0.81** | **0.70** | **0.60** | **0.70** | **0.76** | **0.56** | **0.69** | **0.67** | **0.72** | **0.58** | **0.57** | **0.62** | **0.72** | **0.46** | **0.45** | **0.54** |

Table 13: **3000 editing results for COUNTERFACT.** $T$: Num Edits.

| Method | COUNTERFACT-3000 | | | | | | | | | | | | | | | |
| | $T = 1500$ | | | | $T = 2000$ | | | | $T = 2500$ | | | | $T = 3000$ | | | |
| | Rel. | Gen. | Loc. | Avg. | Rel. | Gen. | Loc. | Avg. | Rel. | Gen. | Loc. | Avg. | Rel. | Gen. | Loc. | Avg. |
| Mistral-7B | | | | | | | | | | | | | | | | |
| **MEMIT** | 0.19 | 0.06 | 0.05 | 0.10 | 0.15 | 0.03 | 0.03 | 0.07 | 0.12 | 0.02 | 0.01 | 0.05 | 0.10 | 0.02 | 0.01 | 0.04 |
| **+O-Edit** | 0.51 | 0.33 | 0.18 | 0.34 | 0.44 | 0.26 | 0.15 | 0.28 | 0.42 | 0.26 | 0.15 | 0.27 | 0.40 | 0.22 | 0.12 | 0.25 |
| **+O-Edit+** | 0.61 | 0.42 | 0.53 | 0.52 | 0.56 | 0.31 | 0.48 | 0.45 | 0.50 | 0.28 | 0.48 | 0.42 | 0.44 | 0.25 | 0.50 | 0.39 |
| **+♠ O-Edit+** | 0.79 | 0.55 | 0.68 | 0.67 | 0.74 | 0.50 | 0.63 | 0.62 | 0.71 | 0.44 | 0.60 | 0.58 | 0.70 | 0.44 | 0.61 | 0.58 |
| Llama3-8B | | | | | | | | | | | | | | | | |
| **MEMIT** | 0.18 | 0.06 | 0.05 | 0.10 | 0.15 | 0.03 | 0.03 | 0.07 | 0.12 | 0.02 | 0.01 | 0.05 | 0.10 | 0.02 | 0.01 | 0.04 |
| **+O-Edit** | 0.55 | 0.40 | 0.27 | 0.41 | 0.46 | 0.30 | 0.25 | 0.33 | 0.42 | 0.28 | 0.24 | 0.31 | 0.39 | 0.30 | 0.20 | 0.26 |
| **+O-Edit+** | 0.79 | 0.44 | 0.28 | 0.50 | 0.65 | 0.40 | 0.26 | 0.43 | 0.54 | 0.33 | 0.24 | 0.37 | 0.46 | 0.29 | 0.22 | 0.32 |
| **+♠ O-Edit+** | 0.91 | 0.45 | 0.56 | 0.64 | 0.86 | 0.41 | 0.51 | 0.59 | 0.82 | 0.40 | 0.47 | 0.56 | 0.80 | 0.36 | 0.44 | 0.53 |

