# OpenReview forum: "O-Edit: Orthogonal Subspace Editing for Language Model Sequential Editing"
_ICLR.cc/2025/Conference — Submitted to ICLR 2025_

### Official Review · Reviewer_qFpa · 2024-10-30

**Soundness:** 2
**Presentation:** 2
**Contribution:** 3
**Rating:** 6
**Confidence:** 4

**Summary:**

This paper proposes a novel approach to continuous editing by orthogonalizing the direction of each knowledge update, thereby minimizing interference between successive updates and reducing the impact of new updates on unrelated knowledge. This results in performance improvements while effectively maintaining the model’s performance on downstream tasks.

**Strengths:**

- This paper proposed an innovative knowledge editing approach to address the challenge of continuous editing.
- Continuous editing of the model is achieved without introducing additional parameters.
- The theoretical derivations are detailed and convincing.

**Weaknesses:**

- The primary experiment was conducted using only CounterFact dataset, without performing tests on a broader set of datasets.
- There is no further elaboration on the time and computational costs associated with O-Edit.
- It would be beneficial to analyze and compare the advantages of O-Edit over other knowledge editing methods such as WiSE and GRACE.
- I noticed that generalization is reduced at T=200. I think this is due to your constraints leading to insufficient thoroughness in editing, which increases generation and decreases localization.
- The theoretical validity of the CGS needs further exploration, such as investigating the relationship between CGS and the offset of the model's hidden state.

**Questions:**

- I found that as the number of edits increases, the loc metric significantly decreases. Is this because the 7B model has a smaller dimensionality? Could you investigate the relationship between O-Edit's loc and the number of edits with different model sizes?
- Additionally, what are your thoughts on the fact that different vectors are almost orthogonal at higher dimensions? Does this mean that the effect is not significant in larger models?
- For methods like ROME and MEMIT that modify model parameters, they usually have better portability. Could you explore how the portability of your method changes after multiple edits?

In summary, this paper presents a significant advancement in sequential knowledge editing by proposing an orthogonal subspace approach. However, expanding the evaluation to more datasets, detailing computational costs, and exploring broader theoretical insights would enhance the paper’s depth and practical relevance. Additionally, addressing the questions raised could provide valuable insights for future work. Overall, this is a promising direction, deserving further investigation.

---

> ### Author Response · Authors · 2024-11-18
>
> Dear Reviewer,
>
>
> We are grateful for your thoughtful feedback and for highlighting key areas of our work. Your insights are invaluable, and we are eager to address your concerns and provide further clarification.
>
> To date, we have evaluated our methods and baselines on three additional knowledge editing datasets. These new datasets differ from the two originally mentioned in the paper by presenting increased editing difficulty and incorporating additional evaluation metrics such as multi-hop ability and portability. Furthermore, we have extended the original 1,500 edits to 3,000 to explore the limits of the orthogonal editing method.
>
> Additionally, we have constructed six different analytical experiments to investigate their impact and mechanisms. These experiments will provide detailed answers to the questions you raised, such as exploring the relationship between CGS and the offset of the model's hidden state, and the relationship between orthogonality and editing performance.
>
> We plan to re-upload the revised paper in two days, which will include improved experimental and analytical details. We will also respond to all your questions individually. We hope this will help you better understand O-Edit and O-Edit+ and our insights into the field of sequential editing. Thank you for your continued attention to our work!

---

> > ### Author Response · Authors · 2024-11-21
> > **paper re-upload**
> >
> > Thank you for your thorough review and insightful comments. In response to your feedback, we have substantially revised our manuscript, particularly strengthening the experimental section. Our major improvements include:
> >
> > 1.We have expanded the evaluation of portability metrics in COUNTERFACT and extended it to 3,000 edits. ﻿
> >
> > 2.We have extended the evaluation to include two additional evaluation datasets, RECENT and WIKICF, as well as four additional evaluation metrics. ﻿
> >
> > 3.We have investigated how different editing methods retain knowledge from different editing stages.
> >
> > 4.We have explored the changes in L2_norm. ﻿
> >
> > 5.We have investigated the editing effects of the editing method on specific types of semantics. ﻿
> >
> > 6.We have also studied the impact of different editing methods on model outputs from two aspects and visualized them using t-SNE dimensionality reduction. ﻿
> >
> > 7.Finally, we have discussed the computational cost of the orthogonal method.
> >
> > We believe these supplementary experiments substantially address the valuable questions you have raised.

---

> ### Author Response · Authors · 2024-11-21
> **QUESTION ANSWER**
>
> **Q1：The primary experiment was conducted using only CounterFact dataset, without performing tests on a broader set of datasets.**
>  (1)To demonstrate the effectiveness of our method, we incorporated additional editing datasets: RECENT and WIKICF. RECENT contains approximately 1,200 recently updated knowledge entries [1], while WIKICF presents more challenging editing targets. Following Zhang et al.[1], we evaluated multiple metrics including Portability (Port.), Subject Aliasing (Alg.), Compositionality and Reasoning (Res.), Forgetfulness (Fog.), and Logical Generalization (Lgn.). The experimental results are presented in Figure 4 on page 7 and Tables 10, 11, and 12 on page 28. (2) Our method demonstrates superior performance across all these additional evaluations. Notably, in RECENT-1200, our method outperforms the baseline by orders of magnitude, while in WIKICF-400, it achieves 5-10 times better performance compared to the original methods.
> (2) We attempted to extend the number of edits to 3000  on COUNTERFACT to explore the limits of this orthogonal editing method. As shown in Table 13 on page 30, we found that at 3000 edits, the orthogonal method could still complete about 50\% of knowledge edits, while traditional methods only achieved 10\%. Moreover, traditional methods seemed to only memorize this 10\% of edited knowledge, almost completely forgetting unrelated knowledge. However, increasing the amount of edited knowledge affects model performance in two ways:  1. New knowledge will overwrite existing knowledge, preventing the model from utilizing original knowledge to solve un-updated downstream tasks. 2.Due to the limited knowledge capacity of a single matrix and the fact that new knowledge affects surrounding knowledge to some extent, this often leads to model parameter and knowledge drift. Although the orthogonal method has largely controlled the model's forgetting of original knowledge and parameter drift, some forgetting of original knowledge is inevitable.
>
>
> **Q2:There is no further elaboration on the time and computational costs associated with O-Edit.**
> A2: We have re-examined the computational aspects of the orthogonal editing method. The additional computation time is primarily due to the singular value decomposition (SVD) of matrices. For $\nabla G$, its SVD is computed once prior to the first edit, with the $V$ matrix saved for reuse. However, for $\Delta W_{\text{total}}$, which is dynamically updated, the SVD must be recomputed after each knowledge edit. On average, computing the SVD for a matrix $W \in \mathbb{R}^{4096 \times 14336}$ takes approximately **4 seconds**, while a single edit using ROME or MEMIT takes around **6 seconds**. For sequential editing, O-Edit and O-Edit+ require only one SVD computation on average per edit, with results significantly surpassing those of traditional methods by several times. While this introduces additional computational overhead primarily from SVD operations, we believe the substantial performance improvements justify these computational costs. The additional computation time is primarily due to the singular value decomposition (SVD) of matrices, but the enhanced editing quality and reduced interference make this trade-off worthwhile.
>
> Table: Computation Time (seconds)
> | Method | COUNTERFACT-1500 | ZsRE-1500 | RECENT-1200 | WIKICF-400 |
> |--------|-----------------|------------|-------------|------------|
> | ROME | 8716 | 8694 | 6917 | 2251 |
> | +O-Edit | 13289 | 13961 | 10663 | 4591 |
> | +O-Edit+ | 12286 | 12664 | 9451 | 4256 |
> | MEMIT | 9122 | 9345 | 7533 | 2614 |
> | +O-Edit | 15438 | 15957 | 12640 | 4997 |
> | +O-Edit+ | 14766 | 14664 | 10854 | 4651 |

---

> ### Author Response · Authors · 2024-11-21
> **QUESTION ANSWER**
>
> **Q3 It would be beneficial to analyze and compare the advantages of O-Edit over other knowledge editing methods such as WISE and GRACE.**
>
> While we have discussed the advantages and working principles of SERACE, WISE, and GRACE in the Related Work section, these methods fundamentally differ from our approach.
>
> (1).SERACE and similar methods do not directly modify model weights; instead, they introduce code-books or external knowledge bases to store new knowledge. To achieve knowledge editing, they train recognizers to identify whether input queries or texts contain new knowledge. When detected, they retrieve and output new answers from the code-book or external knowledge base. However, due to their retrieval-based nature, these methods fail in scenarios requiring reasoning or multi-hop editing, as they do not alter the underlying model. In contrast, our paper focuses on direct weight modification methods like ROME or MEMIT. These approaches aim to directly modify the knowledge embedded within the model, enabling reasonable inference based on the modified knowledge. This superiority is demonstrated by our additional evaluation metrics such as Prot and Res, which showcase the powerful performance of direct parameter modification methods.
>
> (2).Furthermore, direct parameter modification methods enhance our understanding of model mechanisms. For instance, Meng et al. [2] located specific storage positions of knowledge and gender bias through causal tracing, with editing results validating the significance of these locations. Yao et al. [1] conceptualized the model as a graph and identified information flow paths through iterative edge pruning, providing valuable insights for model editing and bias removal. Therefore, direct weight modification methods are intrinsically linked to deepening our understanding of model operational mechanisms and revealing model behaviors.

---

> ### Author Response · Authors · 2024-11-21
> **question answer**
>
> **Q4: I noticed that generalization is reduced at T=200. I think this is due to your constraints leading to insufficient thoroughness in editing, which increases generation and decreases localization.**
>
> You are right! But this is not a limitation of O-Edit and O-Edit+. In this conference (ICLR), several papers [3,4] discussed the limitations of current knowledge editing methods, suggesting that methods like ROME and MEMIT suffer from editing overfitting, where high-probability v vectors cause unrelated sentences to output the edited target token.
>
> To explain this complex issue, let's begin with a hypothesis: Without constraints on the training direction of v, the randomness in gradient descent likely results in v being captured not only by a specific Relation Token (as mentioned in Section 5.3) but by multiple relation tokens. Consequently, whenever v is activated, its information will inevitably be captured by subsequent tokens, transforming the final output to reflect the information contained in v. This explains why ROME and MEMIT demonstrate higher generalization in scenarios with few edits - the trained v tends to predict the target token regardless of the input.
>
> O-Edit and O-Edit+ impose constraints on v training, aiming to create unique influences on subsequent Relation Tokens without affecting others. This results in lower generalization for O-Edit and O-Edit+ in scenarios with few edits. However, in cases requiring over 500 edits, the v trained by conventional methods becomes activated by most relation tokens. Furthermore,  to edit the next knowledge successfully with ROME or MEMIT, the trained v need  to contain more information to counteract the previous effects, creating a vicious cycle that ultimately leads to model collapse. O-Edit and O-Edit+ minimize this impact, enabling successful sequential editing.
>
> To visualize this effect, we investigated the influence of added v on subsequent tokens in Section 5.3 (Further Analysis). The experimental results demonstrate that O-Edit and O-Edit+ minimize the impact on unrelated Relation Tokens, while ROME and MEMIT do not.
>
> **Q5: The theoretical validity of the CGS needs further exploration, such as investigating the relationship between CGS and the offset of the model's hidden state.**
>
> The key contribution of CGS theory lies in its proposition that spaces corresponding to larger singular values represent primary information and knowledge, while spaces associated with lower singular values correspond to irrelevant information and noise. In our experiments, we dynamically select larger singular values to achieve a balance between knowledge editing and knowledge preservation. As shown in Tables 3, 5, and 6, they all demonstrate the relationship between CGS and editing success: larger CGS values better preserve existing knowledge, while smaller CGS values enable more effective editing.
>
> We also investigated the editing success rates at different stages, as illustrated in Figure 3(a), where O-Edit and O-Edit+ demonstrate superior capability in protecting previously edited knowledge, indirectly validating the importance of CGS in preserving original knowledge. Furthermore, in Section 5.3, we discussed the differential impacts of various methods on model outputs. The experimental results confirm that O-Edit and O-Edit+, protected by CGS, exhibit superior properties.

---

> ### Author Response · Authors · 2024-11-21
> **question answer**
>
> **Q6,7: I found that as the number of edits increases, the loc metric significantly decreases. Is this because the 7B model has a smaller dimensionality? Could you investigate the relationship between O-Edit's loc and the number of edits with different model sizes? Additionally, what are your thoughts on the fact that different vectors are almost orthogonal at higher dimensions? Does this mean that the effect is not significant in larger models?**
>
> A7,8: Our initial experiments were conducted on GPT2-XL before transitioning to Llama and Mistral after validating the reliability. Your observation is correct but not complete. The editing performance is influenced not only by model size but also by the editing layer and the L2 norm of the original model weights.
>
> (1) Regarding Orthogonality Impact:
> While Hu et al.[5] suggest that orthogonality between k vectors is the determining factor for the number of model edits, the factors affecting this orthogonality extend beyond model size. Indeed, as models grow larger, the increased dimensionality naturally leads to greater orthogonality in high-dimensional spaces. However, the layer being edited also significantly impacts orthogonality.
> - In the initial layers, where tokens have not yet interacted, the orthogonality between k vectors is maximized
> - As layer depth increases, token information propagates through attention mechanisms, resulting in reduced orthogonality between k vectors in higher layers
> Therefore, to achieve successful sequential editing, it's preferable to target lower layers for editing operations.
>
> (2) Regarding L2 Norm Impact:
> Research by Hu et al.[6] has demonstrated that L2 norm influences the model's continuous editing performance, as verified in Figure 3(c). Additionally, for Llama3-8B, the L2 norm of the initial weights is approximately 78, while for Mistral-7B it is 23. After 1500 edits, Mistral-7B's overall performance is superior to Llama-8B's. In conclusion, we posit that editing performance is influenced by multiple factors beyond model size, including L2 norm, matrix condition number[7], and others.
>
> The experimental results for GPT2-XL are shown in the following table:
>
> | Method | COUNTERFACT-1000 | | |
> |--------|-----------------|--|--|
> |Method| Rel. | Gen. | Loc. |
> | MEMIT| 0.31 | 0.15 | 0.44 |
> | +O-Edit+ | 0.61 | 0.41 | 0.75 |
>
> **Q8: For methods like ROME and MEMIT that modify model parameters, they usually have better portability. Could you explore how the portability of your method changes after multiple edits?**
>
> Q9: We present the portability results in Figure 4 and Table 10, which demonstrate that our orthogonal method significantly outperforms the baseline approaches.
>
> References:
> [1] Yao Y, Zhang N, Xi Z, et al. Knowledge Circuits in Pretrained Transformers[J]. arXiv preprint arXiv:2405.17969, 2024.
> [2] Meng K, Bau D, Andonian A, et al. Locating and editing factual associations in GPT[J]. Advances in Neural Information Processing Systems, 2022, 35: 17359-17372.
> [3] Can Editing LLMs Inject Harm?
> [4] Uncovering Overfitting in Large Language Model Editing.
> [5] Wilke: Wise-layer knowledge editor for lifelong knowledge editing
> [6] Knowledge in superposition: Unveiling the failures of lifelong knowledge editing for large language models
> [7] Prturbation-restrained sequential model editing

---

> > ### Author Response · Authors · 2024-11-24
> >
> > Dear Reviewers,
> > Thank you for your thorough feedback on our manuscript. We have addressed all your comments. With the rebuttal deadline approaching, we would greatly appreciate a discussion regarding our responses. Please let us know if there are any points that require further clarification or additional explanation.
> > Authors

---

> > ### Comment · Reviewer_qFpa · 2024-11-25
> >
> > Dear Authors,
> >
> > Thank you very much for the clarification! Most of them have addressed my concerns.
> >
> > I appreciate the response, especially the newly conducted experiments. Including those in the revised draft will strengthen the paper.
> >
> > Overall, I acknowledge the promising direction of this work, but still have reservations about viewpoints of experiments (such as comparing with more baseline models and discussion on model size), and decide to increase my original scores 5 to 6.
> >
> > BTW, I think there should be some typos, e.g., A7,8 --> A6,7; Q9 --> A8.
> >
> > Best Regards,
> >
> > Reviewer qFpa

---

> ### Author Response · Authors · 2024-11-25
> **question answer**
>
> **Comparing with more baseline and models**
>
>
> Dear Reviewer,
>
> Thank you for your valuable feedback and suggestions on our paper. We have made further additions and improvements based on your comments:
>
> 1.Regarding the batch editing performance of O-Edit: In the latest version, we have supplemented the experiment results showing that O-Edit can maintain around 80% editing accuracy even after 3,000 editing steps, which expands the applicability of O-Edit.
>
> 2.Regarding the comparison between O-Edit and WISE: In Figure 4(d), we compared O-Edit and the SOTA method WISE, which uses additional parameters. The results show that O-Edit+ still exhibits strong competitiveness, while O-Edit outperforms due to its lack of need for additional storage space and inference time.
>
>
> 3.Regarding the supplementation of additional baselines: In the next version's appendix, we will supplement the experimental results of other baseline methods. We want to emphasize that these methods start to degrade model performance around T=200 editing steps [1].
>
> 4.Regarding the supplementation of the GPT2-XL experiment: We have added the GPT2-XL experimental results on page 25, as shown in Figure 9.
>
> **Thank you again for your recognition of our paper and method. We hope these further supplementary experiments can give you a deeper understanding of our method.**
>
> Wishing you all the best with the review process!
>
> [1]Jiang, Houcheng, et al. "Neuron-level sequential editing for large language models." arXiv preprint arXiv:2410.04045 (2024).
>
>
> [2]Mitchell E, Lin C, Bosselut A, et al. Fast model editing at scale[J]. arXiv preprint arXiv:2110.11309, 2021.

---

> ### Author Response · Authors · 2024-11-26
> **Further experiments**
>
> Thank you for your continued attention. We have added some additional experimental content based on the original text.
>
>
>  1. We compared with more baselines, including FT[1], FT-EWC[2], and MEND[3], for better comparison.
>
>
>
>  2. We conducted experiments in four models of the GPT series, small, medium, large, x-large. Our conclusion reveals the scaling law in the field of sequential editing, as well as the model agnostic of our methods. To highlight these two parts, we have modified Table 1 and added Figure 5.
>
>
> 3. Thank you for your attention again!
>
> [1] A comprehensive study of knowledge editing for large language models
>
> [2] WISE: Rethinking the Knowledge Memory for Lifelong Model Editing of Large Language Models
>
> [3]Fast Model Editing at Scale.

---

### Official Review · Reviewer_TtV1 · 2024-10-31

**Soundness:** 2
**Presentation:** 3
**Contribution:** 2
**Rating:** 6
**Confidence:** 4

**Summary:**

The author finds that the update weights of existing knowledge editing methods are within low-rank subspaces. Based on this, the author proposed two knowledge editing methods (O-Edit & O-Edit+) based on orthogonal subspaces to handle sequence editing. O-Edit and O-Edit+ both aim to ensure that the current updates aligns vertically with the previous updates and the original knowledge. O-Edit introduces new loss function to ensure orthogonality, while O-Edit+ directly guarantees orthogonality between different knowledge. The author demonstrates through experiments that ensuring the orthogonality of knowledge helps improve the performance of existing methods in sequence editing.

**Strengths:**

* Logical consistency: based on the assumption that knowledge from orthogonal domains has minimal mutual influence, the author has proposed two methods, O-Edit and O-Edit+, respectively. Furthermore, O-Edit+ ensures a stronger orthogonality between different knowledge. Experimental results also demonstrate that O-Edit+ exhibits superior performance.
* Innovative: the use of orthogonal subspaces to enhance sequence editing is not only novel but also intuitive.
* The ablation study demonstrates that the method proposed in the paper indeed brings performance improvement.

**Weaknesses:**

**Main Weaknesses**

The effectiveness of O-Edit and O-Edit+ in sequence editing needs further validation.

* *W1*: In Line 51, the author says "there is still no effective solution to these problems". However, some progress [1, 2] has been made in sequence editing. I suggest the author compares these methods to demonstrate the effectiveness of O-Edit and O-Edit+ in sequence editing.
* *W2*: I suggest the author to increase the number of sequence edits T, in order to explore the limits of O-Edit and O-Edit+. GRACE [1] can make the number of editing to 3,000.

**Minor Weaknesses**
* *W3*: Previous work [3] has found that increase in the norm of edited parameters leads to a decrease in model performance and edit failures. Therefore, I suggest conducting additional experiments to demonstrate that O-Edit and O-Edit+ can suppress the increase in the weight norm.

**Missing References**
* A Survey on Knowledge Editing of Neural Networks. (2023)
* Editing Large Language Models: Problems, Methods, and Opportunities. (2023)



$Ref$:

[1] Aging with GRACE: Lifelong Model Editing with Discrete Key-Value Adaptors. (2023)

[2] WISE: Rethinking the Knowledge Memory for Lifelong Model Editing of Large Language Models. (2024)

[3] Model Editing at Scale leads to Gradual and Catastrophic Forgetting. (2024)

**Questions:**

**Main Questions**
* *Q1*: There is a research [1] indicating that existing editing methods are not suitable for multi-hop editing. This could be due to conflicts between new and old knowledge. So, I have the following question: can O-Edit and O-Edit+ in this article be applied to multi-hop editing tasks with orthogonal subspace?
* *Q2*: I want to delve deeper into the issue of forgetting previous edited knowledge in O-Edit and O-Edit+. I hope the author studies the issue of forgetting previous edited knowledge when editing up to 1500 pieces of knowledge.

**Minor Questions**
* *Q3*: Do O-Edit and O-Edit+ incur additional time costs?


$Ref$:

[1]  Mquake: Assessing knowledge editing in language models via multi-hop questions. (2023)

---

> ### Author Response · Authors · 2024-11-18
>
> Dear Reviewer,
>
>
> We sincerely appreciate your insightful feedback and for highlighting these critical aspects of our work. Your comments provide us with an invaluable opportunity to address your concerns and offer further clarification.
> To date, we have evaluated our methods and baselines on three additional knowledge editing datasets. These new datasets differ from the two originally mentioned in the paper (COUNTERFACT and ZsRE) by presenting increased editing difficulty and incorporating additional evaluation metrics such as reasoning ability(multi-hop ability) and portability. Furthermore, we have extended the original 1,500 edits to 3,000 to explore the limits of the orthogonal editing method.
> ﻿
>
> To further analyze these two methods, we have conducted six different analytical experiments to investigate their impact and mechanisms. These experiments include examining the norm growth of different editing methods, analyzing the memory and forgetting capabilities of these methods regarding previously edited knowledge, visualizing the impact of the orthogonal method on relational tokens through dimensionality reduction(t-SNE), and evaluating editing performance on special concepts.
> ﻿
>
> We plan to re-upload the revised version within the next two days. In this updated version, we have corrected minor errors from the original, such as changing 'COUNTERFACT' to 'ZsRE' in Table 10 on page 25, and added missing references. Additionally, we will respond to your comments individually. We hope these responses will help you better understand O-Edit and O-Edit+ and our insights into the field of sequential editing. Thank you for your continued attention to our work!

---

> ### Author Response · Authors · 2024-11-21
> **question answer**
>
> **Q1: In Line 51, the author says "there is still no effective solution to these problems". However, some progress [1, 2] has been made in sequence editing. I suggest the author compares these methods to demonstrate the effectiveness of O-Edit and O-Edit+ in sequence editing.**
>
> A1: While we have discussed the advantages and working principles of SERACE, WISE, and GRACE in the Related Work section, our approach fundamentally differs from these methods in both methodology and capabilities.
>
> (1) Methodological Differences and Limitations of External Storage Approaches:
> SERACE and similar methods employ an indirect approach by utilizing external storage mechanisms (code-books or knowledge bases) rather than modifying model weights directly. These methods operate by:
> - Training recognizers to identify queries requiring new knowledge
> - Retrieving and outputting responses from external storage when triggered
>
> However, this retrieval-based architecture has inherent limitations, particularly in scenarios requiring:
> - Complex reasoning capabilities
> - Dynamic knowledge application
>
> In contrast, our approach aligns with direct weight modification methods (like ROME or MEMIT) that alter the model's internal knowledge representations. This enables the model to perform reasoning and inference using the modified knowledge naturally. Our superior performance is quantitatively demonstrated through comprehensive evaluation metrics, including Prot. and Res., which highlight the advantages of direct parameter modification.
>
> (2) Contributions to Model Understanding:
> Beyond practical advantages, direct parameter modification methods provide valuable insights into model mechanisms and behavior:
> - Meng et al.[1] employed causal tracing to precisely locate knowledge and bias storage positions, validating these locations through successful editing results
> - Yao et al.[2] facilitated knowledge editing by conceptualizing models as information flow graphs, using iterative edge pruning to identify critical pathways for knowledge propagation
>
> These findings demonstrate that direct weight modification approaches not only achieve superior editing performance but also contribute significantly to our theoretical understanding of large language models' internal operations and behavioral patterns.
>
> **Q2: I suggest the author to increase the number of sequence edits T, in order to explore the limits of O-Edit and O-Edit+. GRACE [1] can make the number of editing to 3,000.**
>
> A2: We conducted experiments and explanations on 3000 editors in the main text and Table 13. We found that at 3000 edits, the orthogonal method could still complete about 50% of knowledge edits, while traditional methods only achieved 10%. Moreover, traditional methods seemed to only memorize this 10% of edited knowledge, almost completely forgetting unrelated knowledge. However, increasing the amount of edited knowledge affects model performance in two ways:
>
> (1) New knowledge will overwrite existing knowledge, preventing the model from utilizing original knowledge to solve un-updated downstream tasks, as the answers in the downstream task test set remain unchanged.
>
> (2) Due to the limited knowledge capacity of a single matrix and the fact that new knowledge affects surrounding knowledge to some extent, as discussed in the paper's generalization (Gen.), this often leads to model parameter and knowledge drift. Although the orthogonal method has largely controlled the model's forgetting of original knowledge and parameter drift (see section 5.3 Further Analysis), some forgetting of original knowledge is inevitable. For our proposed orthogonal method, the significance of orthogonality lies in minimizing the impact of current knowledge edits on previous knowledge as much as possible.
>
> **Q3：Previous work has found that increase in the norm of edited parameters leads to a decrease in model performance and edit failures. Therefore, I suggest conducting additional experiments to demonstrate that O-Edit and O-Edit+ can suppress the increase in the weight norm.**
>
>
> (1) We conducted additional experiments and presented the results in Figure 3 (c), For the unconstrained method, MEMIT exhibits a high growth trend in the L2 norm. In contrast, the orthogonal method reduces the growth trend of the matrix by constraining the model's update direction.
>
>
> (2) With the explanation provided in Figure 6(d): completing an edit requires a larger activation value to counteract the influence of previous edits, resulting in a vicious cycle and ultimately poor sequential editing performance. In contrast, the activation values for ||ΔW<j · kj||2 (||ΔWunrelated · kj||2) in O-Edit and O-Edit+ remain consistently low, indicating that a large activation value for ||ΔWown · kj||2 is not necessary to complete a new edit.

---

> ### Author Response · Authors · 2024-11-21
> **question answer**
>
> **Q4:Missing References**
>
> We appreciate the reviewer's suggestion and have added the Missing References！
>
> **Q5: There is a research indicating that existing editing methods are not suitable for multi-hop editing. This could be due to conflicts between new and old knowledge. So, I have the following question: can O-Edit and O-Edit+ in this article be applied to multi-hop editing tasks with orthogonal subspace?**
>
> A5: To demonstrate the reasoning of our method, we incorporated additional editing datasets: RECENT and WIKICF. Following Zhang et al.[3], we evaluated multiple metrics including Portability (Port.), Subject Aliasing (Alg.), Compositionality and Reasoning (Res.), Forgetfulness (Fog.), and Logical Generalization (Lgn.). The experimental results are presented in Figure 4 on page 7 and Tables 10, 11, and 12 on page 28.
>
>
>
> **Q6: I want to delve deeper into the issue of forgetting previous edited knowledge in O-Edit and O-Edit+. I hope the author studies the issue of forgetting previous edited knowledge when editing up to 1500 pieces of knowledge.**
>
>
>  A6:We added additional experiments to investigate this issue. We divided the editing process into 15 stages according to the sequence of edits, and evaluated the model after 1500 edits at each stage. As shown in Figure 3(a), the original method MEMIT exhibits a complete forgetting effect on the initial edits. In contrast, O-Edit shows significant improvements compared to MEMIT. Moreover, O-Edit performs best for edits between 1000 and 1500, demonstrating its ability to effectively retain recently edited knowledge. As for O-Edit+, it presents a balanced editing performance, excelling at updating both the initially edited knowledge and the recently edited knowledge.
>
> **Q7: Do O-Edit and O-Edit+ incur additional time costs?**
>
>
> A7:We have re-examined the computational aspects of the orthogonal editing method. The additional computation time is primarily due to the singular value decomposition (SVD) of matrices. For $\nabla G$, its SVD is computed once prior to the first edit, with the $V$ matrix saved for reuse. However, for $\Delta W_{\text{total}}$, which is dynamically updated, the SVD must be recomputed after each knowledge edit. On average, computing the SVD for a matrix $W \in \mathbb{R}^{4096 \times 14336}$ takes approximately **4 seconds**, while a single edit using ROME or MEMIT takes around **6 seconds**. For sequential editing, O-Edit and O-Edit+ require only one SVD computation on average per edit, with results significantly surpassing those of traditional methods by several times. While this introduces additional computational overhead primarily from SVD operations, we believe the substantial performance improvements justify these computational costs. The additional computation time is primarily due to the singular value decomposition (SVD) of matrices, but the enhanced editing quality and reduced interference make this trade-off worthwhile.
>
> Table: Computation Time (seconds)
> | Method | COUNTERFACT-1500 | ZsRE-1500 | RECENT-1200 | WIKICF-400 |
> |--------|-----------------|------------|-------------|------------|
> | ROME | 8716 | 8694 | 6917 | 2251 |
> | +O-Edit | 13289 | 13961 | 10663 | 4591 |
> | +O-Edit+ | 12286 | 12664 | 9451 | 4256 |
> | MEMIT | 9122 | 9345 | 7533 | 2614 |
> | +O-Edit | 15438 | 15957 | 12640 | 4997 |
> | +O-Edit+ | 14766 | 14664 | 10854 | 4651 |
>
> **References**
>
> [1] Meng K, Bau D, Andonian A, et al. Locating and editing factual associations in GPT[J]. Advances in Neural Information Processing Systems, 2022, 35: 17359-17372.
>
> [2]Yao Y, Zhang N, Xi Z, et al. Knowledge Circuits in Pretrained Transformers[J]. arXiv preprint arXiv:2405.17969, 2024.
>
>
> [3]Zhang N, Yao Y, Tian B, et al. A comprehensive study of knowledge editing for large language models[J]. arXiv preprint arXiv:2401.01286, 2024.

---

> > ### Author Response · Authors · 2024-11-21
> > **paper re-upload**
> >
> > we have substantially revised our manuscript, particularly strengthening the experimental section. Our major improvements include:
> >
> > 1.We have expanded the evaluation of portability metrics in COUNTERFACT and extended it to 3,000 edits. ﻿
> >
> > 2.We have extended the evaluation to include two additional evaluation datasets, RECENT and WIKICF, as well as four additional evaluation metrics. ﻿
> >
> > 3.We have investigated how different editing methods retain knowledge from different editing stages.
> >
> > 4.We have explored the changes in L2_norm. ﻿
> >
> > 5.We have investigated the editing effects of the editing method on specific types of semantics. ﻿
> >
> > 6.We have also studied the impact of different editing methods on model outputs from two aspects and visualized them using t-SNE dimensionality reduction. ﻿
> >
> > 7.We have discussed the computational cost of the orthogonal method.
> >
> > 8.Missing References.

---

> ### Comment · Reviewer_TtV1 · 2024-11-22
> **Response to Submission2954 Authors**
>
> Tanks for your reply! Some of my concern has been addressed. However, the author's explanation of memory-based approaches does not convince me. Indeed, memory-based method may have some drawbacks, but I believe it is still an effective approach to knowledge editing. Although the author expands the number of editing iterations to 3,000, the results seem to have less convincing power. At this point, I would think: why don't I use some extra storage space to achieve better performance? There is another paper similar to your method [1], which I didn't mention in the initial response because it is during the ICLR submission period, and the author doesn't need to do the corresponding comparative experiments in the subsequent responses. Despite they only setting the number of edits to 1,000 in the main experiment, this does not affect the persuasiveness of the article, because they are finally able to increase the number of edits to 10,000. I believe that it can illustrate one of the disadvantages of memory-based method (unbearable space consumption). In summary, I really like your method. I think orthogonal subspaces are beautiful. However, I find the effectiveness of O-Edit and O-Edit+ less convincing, which has affected my score.
>
>
> $Ref$:
>
> [1] Reasons and Solutions for the Decline in Model Performance after Editing. (NeurIPS 2024)

---

> ### Author Response · Authors · 2024-11-22
> **question answer**
>
> **Q1 compaerd with current editing methods**
>
> A1: Thank you for your response. We feel helpless and upset. In fact, we tried D4S's approach to solve sequential editing problems in March this year, which we called KKADD (Additional Keys) at the time. However, we abandoned this direction because the results were not as good as O-Edit and O-Edit+. We are very confident in our method's effectiveness. While D4S's method works, it is not as effective as ours for the following reasons:
> ﻿
> 1. D4S [1] uses different evaluation criteria from ours. D4S considers batch editing of mutiple knowledge items at once rather than editing one knowledge once a time (which is difficult to notice for those who haven't conducted editing experiments). In this case, 1000 edits actually equate to only several iterations of editing.
> ﻿
> 2. The evaluation frameworks differ. The paper doesn't specify their evaluation framework and hasn't open-sourced their code. Similarly, as shown in Table 2, in our experiments using the EasyEdit framework, MEMIT achieves an editing accuracy of 0.5 at 500 edits, while D4S (D4S Table 2) only achieves 0.02, which is an unreasonable result.
> ﻿
> 3. Different layer selections: In our experiments, we only discuss editing results for one layer (D4S uses 5 layers) and edit only one piece of knowledge at one time. Both factors increase editing difficulty, yet we try to study the limits of sequential editing (but this was interpreted as poor performance...).
> ﻿
> 4. D4S discusses L1 norm changes (D4S Fig 6). In fact, L1 norm cannot effectively measure matrix properties as it considers the positive/negative nature of each value. We previously measured L1 norm changes in editing matrices and found minimal changes. However, L1 norm and L2 norm aren't directly related - L2 norm might show significant changes even when L1 norm doesn't (O-Edit Figure 3c).
> ﻿
> 5. We spoke with AlphaEdit's authors about a month ago and planned collaboration. We discussed D4S's method and agreed that while it performs well for 200-500 edits, but its Loc. performance drops below baseline methods when edits exceed 1000 (in our settings).
> ﻿
> 6. Additionally, in the AlphaEdit paper (also submitted to ICLR2025), they similarly use batches of 100 edits per iteration, iterating 20-30 times. This leads to superior editing effects because MEMIT can do well for batch knowledge editing.
> ﻿
> 7. This explains why neither AlphaEdit nor D4S discuss improvements to ROME and related experiments, as ROME only suits single knowledge editing, which would show poor results in paper.
> ﻿
> 8. We conducted additional unified experiments to help you compare different editing methods. We selected O-Edit+, AlphaEdit, and D4S. Our experimental setup: Llama3-8B Layer 8, editing one knowledge at one time, under the EasyEdit framework. Other hyperparameter settings showed consistent,  result as shown in the following table:
>
> ﻿| Method | T=200 |  |  |  | T=500 |  |  |  | T=1000 |  |  |  | T=1500 |  |  |  |
> |---|---|---|---|---|---|---|---|---|---|---|---|---|---|---|---|---|
> |  | Rel. | Gen. | Loc. | Avg. | Rel. | Gen. | Loc. | Avg. | Rel. | Gen. | Loc. | Avg. | Rel. | Gen. | Loc. | Avg. |
> | **Llama3-8B** |  |  |  |  |  |  |  |  |  |  |  |  |  |  |  |  |
> | **MEMIT** | 0.85 | 0.51 | 0.22 | 0.52 | 0.50 | 0.35 | 0.10 | 0.32 | 0.28 | 0.10 | 0.05 | 0.14 | 0.18 | 0.06 | 0.05 | 0.10 |
> | &nbsp;&nbsp;**+D4S** | **0.98** | 0.63 | 0.52 | 0.71 | **0.94** | **0.60** | 0.20 | 0.58 | 0.85 | 0.50 | 0.07 | 0.47 | 0.73 | 0.48 | 0.04 | 0.41 |
> | &nbsp;&nbsp;**+Alpha-Edit** | 0.90 | 0.41 | 0.77 | 0.70 | 0.90 | 0.38 | 0.33 | 0.53 | **0.85** | 0.32 | 0.27 | 0.48 | **0.81** | 0.31 | 0.20 | 0.44 |
> | &nbsp;&nbsp;**+O-Edit+** | 0.88 | 0.53 | 0.76 | **0.72** | 0.84 | 0.51 | **0.45** | **0.60** | 0.81 | **0.50** | **0.31** | **0.54** | 0.79 | 0.44 | **0.28** | **0.50** |
>
> Our method showed the best performance, while D4S failed after 1000 edits. In fact, the advantage of D4S method is that it can protect previously edited knowledge as much as possible. However, its ability to forget irrelevant knowledge is stronger than the original method.
>
> Due to the time limit of rebuttal, we will further supplement the effects on batch editing (T=100) in the future to expand to 10000 edits. Finally, we have sufficient confidence in our editing method, and we believe that under the conditions of AlphaEdit or D4S, our method's performance is still optimal. This is due to our deep understanding of orthogonal editing methods.
>
> **Q2 Further Understanding**
>
> This could be two different paths. In our opinion, although the method of directly modifying model weights is inferior to the method of adding additional parameters, the ultimate goal of this method is to find a more effective fine-tuning method that can replace the current gradient descent method. For the search of this method, I believe it can further promote our understanding of the internal operating mechanism of the model.

---

> ### Author Response · Authors · 2024-11-22
>
> [1]Reasons and Solutions for the Decline in Model Performance after Editing. (2024)
>
> [2] AlphaEdit: Null-Space Constrained Knowledge Editing for Language Models
>
> [3] https://github.com/zjunlp/EasyEdit

---

> ### Comment · Reviewer_TtV1 · 2024-11-22
> **Response to Submission2954 Authors**
>
> If I caused any misunderstanding, I would like to apologize. What I want to express is: I think that under sequence editing, 3,000 times doesn't seem to be a very convincing number of edits to exclude the memory-base method.  With the current LLM context window being able to reach 32K, we could even use these 3,000 samples as context to update the knowledge inside the model. The existing memory-base methods can also handle 3,000 samples. Additionally, as far as I know, the MLP's shape of Llama3-8B is 4,096*14,336, so 3,000 edits seem to be less than the rank of MLP. As I mentioned before, you don't need to compare with D4S, but I hope you can clarify the advantages of your method over the previous memory-base method.
>
> I also don't deny the interpretability impact brought by the locate&edit method. However, some experimental results also indicate that the current methods are not accurate in localization the editing layer [1]. The memory-based method can also enhance our understanding of the internal knowledge of the model [2].  Additionally, the author claims that the memory-based approach is not suitable for dynamic knowledge application. I hope to see experimental results demonstrating that O-Edit and O-Edit+ can replace the memory-base approach in such scenarios.
>
> I really appreciate your method, and I hope that this method of keeping knowledge orthogonal can have a far-reaching impact. However, I think the current results are not persuasive enough for me to evaluate the merits and demerits of locate&edit, memory-base and even meta-learning. I believe it's inappropriate not to consider the memory-base method when it's not possible to highlight its serious shortcomings.
>
> If my concern in this regard is resolved, I promise I will improve my scores.
>
>
> $Ref$:
>
> [1] Does Localization Inform Editing? Surprising Differences in Causality-Based Localization vs. Knowledge Editing in Language Models. (NeurIPS 2023)
>
> [2] WISE: Rethinking the Knowledge Memory for Lifelong Model Editing of Large Language Models. (NeurIPS 2024)

---

> ### Author Response · Authors · 2024-11-22
> **question answer**
>
> Thank you for your continued attention!
> ﻿
> **Compare with Memory-based Methods**
> ﻿
> 1. These methods correspond to three cognitive phases of the model, as described by Zhang et al.[1]:
> - **Recognition Phase**: SERAC provides sentences demonstrating factual updates as examples, allowing initial recognition of knowledge to be edited.
> - **Association Phase**: Methods like WISE and GRACE combine or substitute output/intermediate output with learned knowledge representations, similar to how humans connect new ideas to existing concepts.
> - **Mastery Phase**: Methods like ROME and MEMIT directly modify model weights, enabling the model to handle tasks independently without external assistance or merging.
> ﻿
> 2. Memory-based methods require additional computational and storage costs. They can be categorized into two types:
> - **Fixed Storage Cost**: Methods like SERAC[4] and WISE[2] use fixed-size BERT_EXTRA or MLP as external knowledge bases. Balancing knowledge becomes challenging when retrieval needs exceed storage capacity. O-Edit could be one solution to this balance issue. Additionally, retrieval operations incur fixed extra query and inference time.
> - **Variable Storage Cost**: Methods like GRACE[3] maintain an expanding codebook for new knowledge, requiring increased storage space with each edit. With O(N) retrieval complexity, inference time grows with the editing stream, as noted in GRACE and WISE papers.
> ﻿
> 3. These methods typically require fixed input formats and single retrieval operations for answers. However, in practical scenarios, language models need continuous token generation. Performing retrieval for each token generation would significantly increase computational burden.
> ﻿
> 4. Some metrics cannot be compared with localized editing:
> - Zhang et al. provided detailed comparisons in their paper (Page 16 Table 4)
> - These methods struggle with effective subsequent token generation (Section 4.5 Error and Case Analysis)
> ﻿
> 5. Limited Applications:
> - Neither Zhang et al.[1] nor SERAC/WISE/GRACE papers tested these memory-based methods on downstream tasks.
> - These retrieval-based methods fundamentally don't support downstream task evaluation or inference at code level[1, 5].
> - They function more as rigid knowledge retrievers triggered by specific mechanisms rather than versatile language models
> - Localized editing methods attempt to understand and modify internal model mechanisms without additional retrieval time, while measuring downstream task impacts
> ﻿
> 6. O-Edit and other localized editing(ROME,MEMIT) methods successfully address all these challenges.
> ﻿
> ﻿
> References:
> [1] A Comprehensive Study of Knowledge Editing for Large Language Models
> [2] WISE: Rethinking the Knowledge Memory for Lifelong Model Editing of Large Language Models (NeurIPS 2024)
> [3] Aging with GRACE: Lifelong Model Editing with Discrete Key-Value Adaptors
> [4] Memory-based Model Editing at Scale
> [5] https://github.com/zjunlp/EasyEdit

---

> ### Author Response · Authors · 2024-11-23
> **Further Experiments**
>
> Dear Reviewer,
> ﻿
>
> I apologize for my offense yesterday. We have supplemented our paper with comparisons to methods that add extra parameters. We conducted comparisons on the ZsRE dataset at the same editing position (Layer 8) and included the discussion in our paper.
> ﻿
>
> After carefully reviewing the papers and code of WISE and GRACE, we found their editing positions differ from ours. For methods like ROME and MEMIT, editing performance tends to be better in lower layers because the hidden states there have greater orthogonality, resulting in less interference with unrelated knowledge. However, WISE, GRACE, and T-patcher[1] tend to edit higher layers, possibly because they focus more on subtle differences in sentence semantics.
> ﻿
>
> In our additional experiments with 1,500 edits, WISE showed lower performance in editing accuracy metrics, while maintaining minimal disruption to unrelated knowledge. This difference likely stems from the different editing positions used by these two approaches. Notably, we also attempted 500 edits at a higher layer (Layer 27, as used in WISE's code), but did not achieve the high performance reported in the WISE paper - editing accuracy and generalization remained around 60, while Loc stayed around 85.
> ﻿
>
> In our experimental setup with an A100-40G GPU and Llama3-8B model, methods requiring additional parameters such as WISE were unable to support float32 precision operations. We therefore conducted all experiments using float16 precision, where WISE consumed approximately 21GB of GPU memory, while O-Edit+ demonstrated more efficient memory usage at around 17GB.
>
>
> We sincerely apologize again for yesterday's offense and hope that this additional content helps you better understand the distinctions between these two methods.
> ﻿
>
> [1] Transformer-Patcher: One Mistake Worth One Neuron

---

> > ### Author Response · Authors · 2024-11-24
> >
> > Dear Reviewers,
> > Thank you for your thorough feedback on our manuscript. We have addressed all your comments. With the rebuttal deadline approaching, we would greatly appreciate a discussion regarding our responses. Please let us know if there are any points that require further clarification or additional explanation.
> > Authors

---

> ### Comment · Reviewer_TtV1 · 2024-11-25
> **Response to Submission2954 Authors**
>
> I appreciate the authors' effort, and I have decided to improve my score to 6. However, I have two additional questions that I hope the authors will resolve in the future:
>
> a) Do all knowledge have to maintain orthogonality? When some edited knowledge is related, do we still need to maintain the orthogonality of knowledge?
>
> b) I thought of a scenario where the memory-base method is not good at but locate&edit excels at: continually editing the object of (s, r, o).   For instance, with the changes in the world, such as presidential elections, the object of the triplet (Country A, President is, Person B) will constantly change (**Person B -> Person B' -> Person B'' -> ...**).

---

> ### Author Response · Authors · 2024-11-25
> **question answer**
>
> **Do all knowledge have to maintain orthogonality? When some edited knowledge is related, do we still need to maintain the orthogonality of knowledge?**
>
> 1.We also appreciate your further thinking on sequential editing. Orthogonality is a good condition, but its constraints are also evident because the space of the model is limited. However, in current knowledge editing, more consideration is given to triplet type knowledge, and in the code (ROME, MEMIT), it filters out edits to the same topic, meaning that each subject will only be edited once. This greatly reduces the difficulty of editing, as the orthogonality between different tokens is stronger.
>
>
> 2.When the knowledge to be edited is relevant, they do not need to ensure complete orthogonality. And as the editing knowledge becomes more similar, their orthogonality should be smaller. However, there is no good conclusion on how to define the correlation between knowledge. Consider an example: The President of US is. Should this knowledge be similar to 1. The President of the UK, or 2. The highest leader of the United States of America? We humans would definitely consider it more similar to 2. However, when encoding, the model considers it to be more similar to 1, which is why knowledge editors need to consider the Gen metric.
>
> 3. As we have responded to Reviewer 1, we believe that the goal of sequential editing is:
>
> $$\int_0^{+\infty} \frac{\partial f_{\theta +\Delta \theta}(x_{\text{old}})}{ \partial \Delta \theta} d(x_{\text{old}}) = 0$$
>
> . It should be a comprehensive consideration of all parameters of the model. The current method focuses on updating some parameters. We will further discuss the correlation between editing scope and editing effect in our future work, and we aim to extend the O-Edit method to full parameter editing.
>
> **I thought of a scenario where the memory-base method is not good at but locate&edit excels at: continually editing the object of (s, r, o). For instance, with the changes in the world, such as presidential elections, the object of the triplet (Country A, President is, Person B) will constantly change (Person B -> Person B' -> Person B'' -> ...)**
>
> What a wonderful scenario, and we have further thought about it, hoping to develop stronger editing methods.
>
>
> ﻿
> **Finally, we would like to express our sincere gratitude for the effort you have put into reviewing our paper. Through this rebuttal, we have learned how to produce a truly excellent piece of work.**

---

> ### Author Response · Authors · 2024-11-26
> **Further experiments**
>
> Thank you for your continued attention. We have added some additional experimental content based on the original text.
>
>
>  1. We compared with more baselines, including FT[1], FT-EWC[2], and MEND[3], for better comparison.
>
>
>
>  2. We conducted experiments in four models of the GPT series, small, medium, large, xlarge. The key reason why we did not compare on larger models is that different types of models do not have comparability (such as different norms and condition numbers). Our conclusion reveals the scaling law in the field of sequential editing, as well as the model agnostic of our methods. To highlight these two parts, we have modified Table 1 and added Figure 5.
>
>
> 3. Thank you for your attention again!
>
> [1] A comprehensive study of knowledge editing for large language models
>
> [2] WISE: Rethinking the Knowledge Memory for Lifelong Model Editing of Large Language Models
>
> [3]Fast Model Editing at Scale.

---

### Official Review · Reviewer_NXNx · 2024-11-03

**Soundness:** 3
**Presentation:** 3
**Contribution:** 2
**Rating:** 5
**Confidence:** 3

**Summary:**

This paper introduces O-Edit and O-Edit+, two methods for sequential knowledge editing in large language models (LLMs) that address the challenge of catastrophic forgetting during multiple edits. The key idea lies in performing edits in orthogonal subspaces, ensuring that new knowledge updates minimally interfere with both previously edited knowledge and the model's implicit knowledge. The methods work by projecting update directions into orthogonal subspaces and using post-processing techniques to maintain complete orthogonality between different knowledge updates. Through extensive experiments on Mistral-7B and Llama3-8B models using the COUNTERFACT and ZsRE datasets, the authors demonstrate that their approaches significantly outperform existing methods like ROME and MEMIT, especially when handling large numbers of sequential edits (up to 1,500). The methods also better preserve model performance on downstream tasks while requiring minimal additional parameters. The paper provides theoretical and experimental evidence showing that strong orthogonality between update matrices is crucial for successful sequential editing, offering a promising direction for future research in this area.

**Strengths:**

1. The results look excellent compared to previous approaches.
2. The paper is well-written and easy to follow.
3. The analysis is comprehensive, which helps to understand the key insights of the proposed method.

**Weaknesses:**

1. The method is only evaluated on two datasets. It would be interesting to see results on more scenarios where knowledge editing is important.
2. Although the idea of orthogonal subspace editing looks interesting, I am wondering is it possible to ensure the orthogonality of the direction of each knowledge update when the editing number scales to the millions level? In updating a pre-training LLM, I think it is meaningful to investigate knowledge editing where the training corpus is so large that the number of edits is hard to count.

**Questions:**

Have you explored the influences of context lengths on knowledge editing?

---

> ### Author Response · Authors · 2024-11-18
>
> Dear Reviewer,
>
> Thank you for your insightful feedback and for highlighting these important aspects of our work. We appreciate the opportunity to address your concerns and provide further clarification.
>
> To date, we have evaluated our methods on three additional editing datasets, which include longer contextual content and expanded evaluation metrics such as reasoning ability and portability. We also assessed the performance metrics of different editing methods under 3,000 edits. The experimental results consistently highlight the advantages of O-Edit and O-Edit+.
>
> Additionally, we have constructed six different analytical experiments and a sequential editing target formula to better understand the impact and mechanisms of O-Edit and O-Edit+. We plan to upload the additional experimental content and textual analysis within the next two days.
>
> Once we have completed the re-upload of the paper, we will provide a response in a separate comment. We hope these responses will help you better understand O-Edit and O-Edit+ and our insights into the field of sequential editing. Thank you for your continued attention to our work!

---

> ### Author Response · Authors · 2024-11-21
> **Paper re-upload and question answer**
>
> Thank you for your continued attention. We have re-uploaded the manuscript and made extensive updates to the experimental section to help readers gain a deeper understanding of our methodology. The updates to the experimental section specifically include:
>
>
> 1.We have expanded the evaluation of portability metrics in COUNTERFACT and extended it to 3,000 edits.
> ﻿
>
>
> 2.We have extended the evaluation to include two additional evaluation datasets, RECENT and WIKICF, as well as four additional evaluation metrics.
> ﻿
>
>
> 3.We have investigated how different editing methods retain knowledge from different editing stages.
>
>
>
> 4.We have explored the changes in L2_norm.
> ﻿
>
>
> 5.We have investigated the editing effects of the editing method on specific types of semantics.
> ﻿
>
> 6.We have also studied the impact of different editing methods on model outputs from two aspects and visualized them using t-SNE dimensionality reduction.
> ﻿
>
>
> 7.Finally, we have discussed the computational cost of the orthogonal method.

---

> ### Author Response · Authors · 2024-11-21
> **quesition answer**
>
> Q1：The method is only evaluated on two datasets. It would be interesting to see results on more scenarios where knowledge editing is important.
>
> A1: To demonstrate the effectiveness of our method, we incorporated additional editing datasets: RECENT and WIKICF. Following Zhang et al.[1], we evaluated multiple metrics including Portability (Port.), Subject Aliasing (Alg.), Compositionality and Reasoning (Res.), Forgetfulness (Fog.), and Logical Generalization (Lgn.). The experimental results are presented in Figure 4 on page 7 and Tables 10, 11, and 12 on page 28. Our method demonstrates superior performance across all these additional evaluations. Notably, in RECENT-1200, our method outperforms the baseline by orders of magnitude, while in WIKICF-400, it achieves 5-10 times better performance compared to the original methods.
>
> Q2：Although the idea of orthogonal subspace editing looks interesting, I am wondering is it possible to ensure the orthogonality of the direction of each knowledge update when the editing number scales to the millions level? In updating a pre-training LLM, I think it is meaningful to investigate knowledge editing where the training corpus is so large that the number of edits is hard to count.
>
> A2: We attempted to extend the number of edits to 3000 to explore the limits of this orthogonal editing method. As shown in Table 13 on page 30, we found that at 3000 edits, the orthogonal method could still complete about 50% of knowledge edits, while traditional methods only achieved 10%. Moreover, traditional methods seemed to only memorize this 10% of edited knowledge, almost completely forgetting unrelated knowledge. However, increasing the amount of edited knowledge affects model performance in two ways:
>
> (1) New knowledge will overwrite existing knowledge, preventing the model from utilizing original knowledge to solve unupdated downstream tasks, as the answers in the downstream task test set remain unchanged.
>
> (2) Due to the limited knowledge capacity of a single matrix and the fact that new knowledge affects surrounding knowledge to some extent, as discussed in the paper's generalization (Gen.), this often leads to model parameter and knowledge drift. Although the orthogonal method has largely controlled the model's forgetting of original knowledge and parameter drift (see section 5.3 Further Analysis), some forgetting of original knowledge is inevitable. For our proposed orthogonal method, the significance of orthogonality lies in minimizing the impact of current knowledge edits on previous knowledge as much as possible. Space within a matrix is limited, and we cannot achieve orthogonality for millions of updates. However, we can control the mutual orthogonality between these pieces of knowledge to reach maximum potential by adjusting hyperparameters (Eq8, Eq10), thereby achieving a balance between preserving various aspects of knowledge.
>
> (3) Furthermore, we have recently been attempting to extend this knowledge editing method, which is limited to single or few matrices, to full parameters, as knowledge does not exist solely within a single matrix but affects the model's final output through information flow [2]. To this end, we propose an optimization criterion for continuous editing: for model $f$ and its parameters $\theta$, regarding the $\Delta \theta$ to be added when updating new knowledge. The added $\Delta \theta$ should satisfy:
>
> $$\int_0^{+\infty} \frac{\partial f_{\theta +\Delta \theta}(x_{\text{old}})}{ \partial \Delta \theta} d(x_{\text{old}}) = 0$$
>
> ROME and MEMIT are approximate solutions to this equation, both constraining this condition to single or few matrices. The true sequential editing objective should target all parameters of the model. Nevertheless, we believe that the introduction of ROME and MEMIT is highly significant, as they solved how to maximally differentiate between the $k_*$ to be edited and the $K$ of unrelated knowledge when updating a single matrix. In our future work, we aim to incorporate this method into LoRA and attempt full-parameter fine-tuning to address the continuous editing problem.

---

> > ### Author Response · Authors · 2024-11-21
> > **question answer**
> >
> > Q3：Have you explored the influences of context lengths on knowledge editing?
> >
> > A3：Following Meng et al.[3], we focus on knowledge in the form of triples $(s,r,o)$. For knowledge with longer content, we address two key aspects:
> >
> > (1) Leading words preceding knowledge may affect the $k_*$ of subject_tokens. We have accounted for this challenge in our experiments, as shown in Equation 16, by utilizing prefix questions when calculating $k_*$ that represents the subject. This approach enhances the robustness of $k_*$, ensuring that the model can accurately identify subject tokens' $k_*$ and activate corresponding $v_*$ even in the presence of extensive preceding text.
> >
> > (2) The object $(o)$ often consists of multiple tokens rather than a single token, meaning the answer is composed of a sequence of tokens. We investigated this multi-token answer scenario using the WIKICF dataset, with results presented in Figure 4 and Table 12. Our findings reveal that all editing methods show suboptimal performance in continuously editing such multi-token answers. This limitation arises because the trained $v_*$ needs to represent information for a group of tokens, which is inherently more challenging than single-token representation. This challenge is one of the primary motivations behind our exploration of full-parameter editing approaches.
> >
> > [1] Zhang, N., Yao, Y., Tian, B., et al. A comprehensive study of knowledge editing for large language models. arXiv preprint arXiv:2401.01286, 2024.
> >
> > [2] Yao, Y., Zhang, N., Xi, Z., et al. Knowledge Circuits in Pretrained Transformers. arXiv preprint arXiv:2405.17969, 2024.
> >
> > [3] Meng, K., Bau, D., Andonian, A., et al. Locating and editing factual associations in GPT. Advances in Neural Information Processing Systems, 35:17359-17372, 2022.

---

> > ### Comment · Reviewer_NXNx · 2024-11-21
> > **Response**
> >
> > Although compared to existing approaches, orthogonal editing could achieve better performance. However, at the level of 3000 edits, only 50% can be completed. I feel that this is far from making this technique useful and practical. For example, one natural scenario I can think of is to make ChatGPT up to date, so we need to edit its outdated knowledge. In this case, I think we will need way more than 3000 edits.
> >
> > In general, I think stronger motivations to demonstrate the significance of the problem and more illustrations about why the proposed technique will be useful could make the paper better.

---

> ### Author Response · Authors · 2024-11-21
> **question answer**
>
> **1. Understanding the Scale of Editing**
>
> 1. We understand your concern, **But research is a gradual process, In this work, we have successfully expanded the effective editing capacity from 100 edits to thousands of edits, which is significant**.  Further, **we discussed the most challenging condition in our paper - editing only one piece of knowledge at a time and repeating this process 1,500 times**, which means 1,500 iterations. However, in practical scenarios, it's unlikely to edit just one piece of knowledge at a time; instead, we might edit 5, 10, or even 1,000 pieces simultaneously. Under these conditions, our method can scale to tens of thousands of edits. **Even if editing can modify one million pieces of knowledge at once and is fully parameterized, it will still significantly disrupt unrelated existing knowledge, and If the current knowledge editing methods can handle millions of edits, then there would be no need for research in this field.** Nevertheless, the limitation of editing a single matrix or few matrices lies in their limited information capacity - for instance, millions of edits would require more model components.
>
> 2. The purpose of knowledge editing is to precisely modify specific knowledge within the model, whereas fine-tuning, as a mainstream method, cannot achieve this goal due to the randomness inherent in gradient descent. Applying existing fine-tuning methods to update knowledge in ChatGPT yields **inferior results** [1,2,3]. Moreover, ChatGPT's larger parameter dimensionality suggests that O-Edit and O-Edit+ can enable more successful edits in larger models.
>
>
> 3. We believe our research is constructive and can guide and influence subsequent studies. **All research is not a one-time effort; addressing a major research direction requires the integration of many studies. If you could spend a little of your valuable time reading the references as others reviewers,  we believe you will gain a deeper understanding of this field and its advancements.**
>
> **2. Motivations**
>
> **But our motivation was already early introduced in the appendix of the article and in Section 5.3**:
>
> As discussed in concurrent papers [5,6], current knowledge editing methods (ROME, MEMIT) suffer from overfitting issues, where the trained v exhibits particularly strong generalization. As evidenced by the Loc (Localization) metric in our editing dataset, ROME and MEMIT demonstrate a tendency to alter predictions for unrelated knowledge. **This behavior indicates that existing editing methods fail to maintain independence between dozens of edits under sequential editing conditions.**
>
> Without constraints on the training direction of v, the randomness in gradient descent likely results in v being captured  **not only by a specific Relation Token**  (as mentioned in Section 5.3), but by multiple relation tokens. Consequently, whenever v is activated, its information will inevitably be captured by subsequent tokens, transforming the final output to reflect the information contained in v. This explains why ROME and MEMIT demonstrate higher generalization in scenarios with few edits - the trained v tends to predict the target token regardless of the input.
>
> Theoretically, an effective editing method should train v to be captured by only one type of relation, not all. To achieve this, we propose training v in orthogonal space. O-Edit and O-Edit+ impose constraints on v training, aiming to create unique influences on subsequent Relation Tokens **without affecting others**.  To visualize this effect, we investigated the influence of added v on subsequent tokens in Section 5.3 (Further Analysis). The experimental results demonstrate that O-Edit and O-Edit+ minimize the impact on unrelated Relation Tokens.
>
> **References:**
> [1] Mitchell, Eric, et al. Memory-Based Model Editing at Scale. ICML 2022.
>
> [2] A comprehensive study of knowledge editing for large language models
>
> [3] De Cao, Nicola, et al. Editing Factual Knowledge in Language Models. EMNLP 2021.
>
> [4] Meng, K., Bau, D., Andonian, A., et al. Locating and editing factual associations in GPT. Advances in Neural Information
> Processing Systems, 35:17359-17372, 2022.
>
> [5] Can Editing LLMs Inject Harm?
>
> [6] Uncovering Overfitting in Large Language Model Editing.

---

> ### Author Response · Authors · 2024-11-25
> **question answer**
>
> Thank you for your continued attention. We have re-re-uploaded the manuscript and made extensive updates to the experimental section to help readers gain a deeper understanding of our methodology. The updates to the experimental section specifically include:
>
> 1.We have also considered the sustained editing performance of O-Edit+ in the case of batch=100, as shown in Tables 1 and 9. In this scenario, O-Edit+ can maintain around 80% editing success rate and 50% generalization, demonstrating its strong capabilities, T=3000.
>
> 2.We have also expanded the editing model to GPT2-XL(page 25), and in this case, O-Edit+ is still able to achieve around 80% editing success rate, while MEMIT falls short at less than half.
>
> 3.We have conducted a comparison between our method and the existing SOTA method WISE[1], and the results have shown that our method exhibits very strong competitiveness. However, the WISE method requires the addition of extra parameters and inference time, as shown on page 7, Figure 4.
>
> [1] WISE: Rethinking the Knowledge Memory for Lifelong Model Editing of Large Language Models. (2024)
>
>
> **Finally, we would like to express our sincere gratitude for your continued engagement and support again. If there are any other areas that you have questions or concerns about, please do not hesitate to let us know. We will make our best efforts to address any remaining confusion or queries you may have.**

---

> ### Author Response · Authors · 2024-11-26
> **Further experiments**
>
> Thank you for your continued attention. We have added some additional experimental content based on the original text.
>
>
>  1. We compared with more baselines, including FT[1], FT-EWC[2], and MEND[3], for better comparison.
>
>
>
>  2. We conducted experiments in four models of the GPT series, small, medium, large, xlarge. The key reason why we did not compare on larger models is that different types of models do not have comparability (such as different norms and condition numbers). Our conclusion reveals the scaling law in the field of sequential editing, as well as the model agnostic of our methods. To highlight these two parts, we have modified Table 1 and added Figure 5.
>
>
> 3. Thank you for your attention again!
>
> [1] A comprehensive study of knowledge editing for large language models
>
> [2] WISE: Rethinking the Knowledge Memory for Lifelong Model Editing of Large Language Models
>
> [3]Fast Model Editing at Scale.

---

### Author Response · Authors · 2024-11-26
**Summary of Revisions**

As the deadline for submitting the revised version approaches, we have summarized the updates made to our work to enable the reviewers and Chairs to reasonably evaluate our contributions. Below are the key improvements and differences compared to the initial submission:
﻿
1. **Expanded Dataset Comparisons**:
- The number of datasets has been increased from 2 to 4.
- Evaluation metrics have been expanded from 3 to 8, including new metrics for inference capability, portability, and forgetting.
﻿
2. **Broader Model Coverage**:
- We extended the range of evaluated models from 0.1B to 8B parameters.
- Scaling laws for continuous editing were revealed, demonstrating the scalability of our proposed method.
﻿
3. **Increased Editing Frequency**:
- The editing frequency was increased to 3000 edits, and experiments with up to 100 knowledge updates per edit were conducted.
﻿
4. **Enhanced Baseline Comparisons**:
- We included additional baselines (FT, FT-EWC, MEND, WISE) for a more comprehensive comparison of various aspects.
﻿
5. **Analysis of Editing Capability Before Forgetting**:
- We investigated how different editing methods retain knowledge from different editing stages. and demonstrated our methods improved performance on all relationships.
﻿
6. **Impact on Matrix L2 Properties**:
- Analyzed how our method affects the L2 norm of matrices during editing.
﻿
7. **Token-Level Analysis**:
- Investigated the performance advantage of the orthogonal method, showing that it minimally impacts subsequent tokens from a relationship token perspective.
﻿
8. **Computation Time Analysis**:
- Evaluated the computational cost of the method, highlighting that computation time decreases as batch size increases.
﻿

**We believe that the final revised version has addressed most of the issues raised by all reviewers.**
﻿


**Acknowledgment**

**We would like to sincerely thank the reviewers, especially TtV1, qFpa and NXNx, for their thoughtful feedback and valuable suggestions, which have greatly improved the quality and clarity of our work.**

**By the way, our paper withdrawal was not due to a lack of confidence in our work or dissatisfaction with the conference, but rather due to an accidental mistake during the check process. We greatly appreciate the chair’s decision to give us another chance after the withdrawal and reversion of withdrawn submission. For this conference and the rebuttal phase, we have made the almost effort we could.**

---

### Note · Authors · 2024-10-02

I have read and agree with the venue's withdrawal policy on behalf of myself and my co-authors.

---

> ### Note · Program_Chairs · 2024-10-05
>
> We approve the reversion of withdrawn submission.

---

### Meta-Review · Area_Chair_VLVh · 2024-12-20

**Metareview:**

This paper presents O-Edit and O-Edit+, two methods for sequential knowledge editing in large language models (LLMs) that mitigate catastrophic forgetting during multiple edits.  The approaches involve projecting update directions into orthogonal subspaces and employing post-processing to maintain strict orthogonality between updates. Extensive experiments on Mistral-7B and Llama3-8B models using the COUNTERFACT and ZsRE datasets show that O-Edit and O-Edit+ outperform baselines like ROME and MEMIT, particularly in handling up to 1,500 sequential edits. Theoretical and experimental results underscore the importance of strong orthogonality in update matrices, highlighting a promising avenue for sequential knowledge editing in LLMs.  While the idea of projecting update directions into orthogonal subspaces is promising, achieving fully orthogonal subspaces within large language models remains a significant technical challenge. Given the rapid advancements in this field, the authors are encouraged to revise the paper thoughtfully based on the reviewers' feedback.

**Additional Comments On Reviewer Discussion:**

During the discussion, the authors provided a substantial amount of additional experiments, and most reviewers found the work interesting.

---

### Decision · Program_Chairs · 2025-01-22

Reject